# Global-correlated 3D-decoupling Transformer for Clothed Avatar Reconstruction

**Zechuan Zhang[1], Li Sun[1], Zongxin Yang[1], Ling Chen[2], Yi Yang[1†]**
[1] ReLER, CCAI, Zhejiang University
[2] AAII, University of Technology Sydney

## Abstract

Reconstructing 3D clothed avatars from single images is a challenging task, especially when encountering complex poses and loose clothing. Current methods exhibit limitations in performance, largely attributable to their dependence on insufficient 2D image features and inconsistent query methods. Owing to this, we present **G**lobal-correlated 3D-decoupling **T**ransformer for clothed **A**vatar reconstruction (**GTA**), a novel transformer-based architecture that reconstructs clothed human avatars from monocular images. Our approach leverages transformer architectures by utilizing a Vision Transformer model as an encoder for capturing global-correlated image features. Subsequently, our innovative 3D-decoupling decoder employs cross-attention to decouple tri-plane features, using learnable embeddings as queries for cross-plane generation. To effectively enhance feature fusion with the tri-plane 3D feature and human body prior, we propose a hybrid prior fusion strategy combining spatial and prior-enhanced queries, leveraging the benefits of spatial localization and human body prior knowledge. Comprehensive experiments on CAPE and THuman2.0 datasets illustrate that our method outperforms state-of-the-art approaches in both geometry and texture reconstruction, exhibiting high robustness to challenging poses and loose clothing, and producing higher-resolution textures. Codes are available at https://github.com/River-Zhang/GTA.

## 1 Introduction

As virtual worlds and metaverse technology gain popularity, the demand for advanced techniques to reconstruct 3D clothed human avatars from single images is rapidly increasing. These techniques [1, 2, 3, 4, 5, 6, 7, 8, 9] are employed across various areas, such as AR/VR, social telepresence, virtual try-on, or the movie industry. However, in-the-wild images often present challenges, such as loose clothing and complex poses, which are not typically found in training data. As a result, there is a pressing need for models that can effectively generalize to these scenarios and reconstruct accurate, animatable, and high-resolution 3D human avatars.

In light of the significant progress made in 3D clothed human avatar reconstruction, existing models still face two main limitations: (i) *Overreliance on 2D image features*. Sole dependence on 2D CNN-based features compromises the accuracy of 3D object reconstructions due to the lack of global correlation. Despite the integration of 3D features from human body priors in methods like [3, 2, 7, 10], their inconsistent performance with loose clothing and challenging poses (See Fig. 2) indicates insufficient integration. Additionally, optimization-based methods [4, 11, 12, 13] can be complex and prone to errors, reducing reliability. (ii) *Inconsistent query methods*. Current strategies for querying features differ and have drawbacks. The pixel-aligned method [1, 6] directly

---

† : the corresponding author.

37th Conference on Neural Information Processing Systems (NeurIPS 2023).

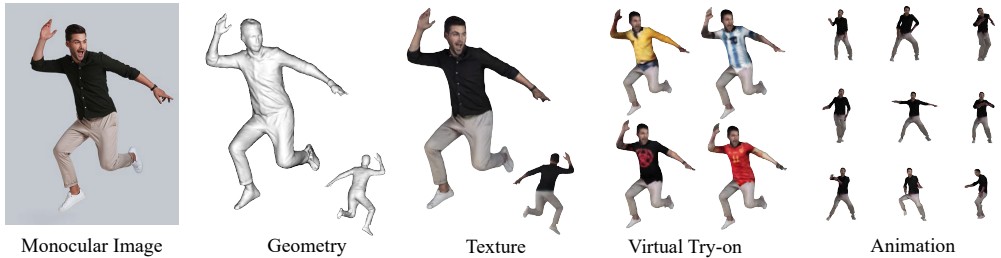

| Monocular Image | Geometry | Texture | Virtual Try-on | Animation |

Figure 1: Given a monocular image as input, GTA reconstructs the full 3D geometry and texture of the subject portrayed, allowing for various applications such as virtual try-on and animation.

projects query points on feature maps but lacks guidance from human body prior, while the prior-guided strategy [7] integrates features on a human body prior but may lead to loss of detailed information on the image and result in 3D avatar reconstructions with an increased level of fuzziness.

Considering the limitations discussed above, we propose that 2D feature maps are insufficient for 3D reconstruction tasks, while global-correlated 3D feature representations offer a more effective solution. Traditional 3D representations are space-intensive and inefficient, necessitating alternatives such as the memory-conserving tri-plane representation [14]. However, generating global-correlated 3D representations from monocular images remains challenging due to difficulties in obtaining orthogonal plane feature maps. Our approach employs learnable embeddings and cross-attention mechanisms to effectively model intricate cross-plane relationships, enabling robust and precise 3D feature extraction. Furthermore, it is important to develop a strategy that synergizes various query methods while maintaining simplicity and efficiency. By combining existing strategies for 3D features, our method leverages localized spatial features and prior knowledge of human body structure, resulting in a balanced feature extraction process that improves reconstruction performance.

In response to the identified challenges, we present **GTA** (**G**lobal-correlated 3D-decoupling **T**ransformer for clothed **A**vatar reconstruction), employing a novel global-correlated 3D-decoupling transformer and a hybrid prior fusion strategy for comprehensive 3D geometry and texture reconstruction. Our vision transformer-based encoder extracts global-correlated features from the input image, while our unique 3D-decoupling decoder disentangles tri-plane 3D features using learnable embeddings as queries. This integration of global-correlated encoding and 3D-decoupling decoding effectively captures the 3D avatar structure from a single image. To further enhance feature fusion, our hybrid prior fusion strategy combines spatial and prior-enhanced queries, leveraging the benefits of spatial localization and human body prior knowledge. This efficient and accurate integration strategy achieves state-of-the-art performance in single-view human avatar reconstruction.

Our proposed model, trained on THuman2.0 [15], outperforms state-of-the-art(SOTA) methods in geometry and texture reconstruction. We achieve a significant reduction in Chamfer distance on CAPE-FP [16] test dataset, below 0.8cm for the first time, and demonstrate superior side-view normal performance, illustrating our method's efficacy in reconstructing accurate 3D clothed human avatars. Our model excels in handling complex poses and loose clothing, and attains state-of-the-art texture reconstruction with higher PSNR scores. Moreover, it can be extended to animation and virtual try-on applications, showcasing its wide-ranging real-world potential. Our main contributions include:

- We introduce a novel global-correlated 3D-decoupling transformer that effectively disentangles tri-plane features, thereby substantially enhancing the reconstruction of clothed avatars from 2D images. To the best of our knowledge, our approach is the pioneering application of transformers in 3D feature decoupling for monocular human avatar reconstruction tasks.

- We put forward an innovative hybrid prior fusion strategy for feature query, combining spatial query's localization capabilities with prior-enhanced query's ability to incorporate knowledge of the human body prior, ultimately leading to improved geometry and texture reconstruction performance.

- Our proposed model achieves state-of-the-art performance in both clothed human geometry and texture reconstruction, outperforming previous models and exhibiting enhanced side-view normal performance.

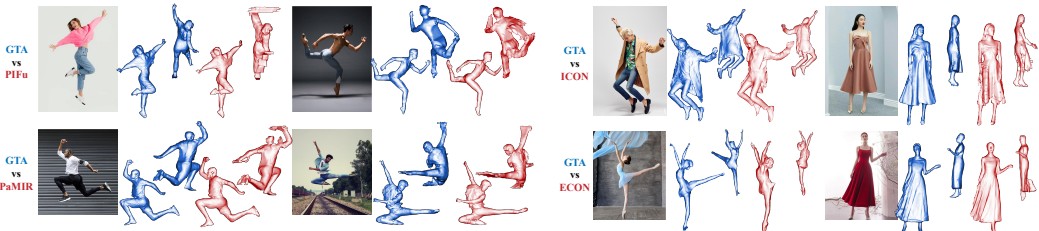

Figure 2: GTA vs. SOTA. SOTA methods (red) are vulnerable to challenging poses and loose clothing, leading to artifacts such as non-human shapes (PIFu [1], PaMIR [3]), incomplete clothing reconstruction (ICON [2]), and erroneous stitching (ECON [4]). GTA deals with these challenges and produces high-quality results (blue).

## 2 Related Work

**Monocular Human Reconstruction** has been an active area of research for many years. This task is inherently ill-posed due to the lack of 3D information, requiring additional assumptions or prior knowledge to recover the full 3D structure. Previous research has proposed effective parametric human prior models [17, 18, 19, 20, 21], which employs statistical methods to reduce the variations in human body shape and pose to a compact set of parameters. By leveraging this model, subsequent research has proposed novel methods to estimate or regress the model parameters from a single RGB image [22, 23, 24, 25, 26]. However, the human prior models can only capture a minimally clothed body without complex details like garments, adornments, or hairstyles. To address this limitation, some researchers add offsets on the top of prior body vertices to simulate outfits [27, 28, 29, 30, 31]. While these methods can effectively represent clothing close to the body surface and use blending weights of surface vertices to drive the clothing, they are not suitable for geometry topology far from the human body, such as robes and dresses.

In order to overcome the constraints imposed on reconstruction by clothing shape and type, researchers have explored various alternative representations for the human body, including voxels [32, 33], visual hulls [34], double depth maps [11, 12, 13, 4], and UV maps translation [35]. Among these diverse methods, implicit function-based methods [36] have shown the most remarkable performance. Saito et al. introduced PIFu [1], which firstly incorporates implicit functions into the problem of human body reconstruction. The method leverages a CNN-based neural network to extract features from 2D images and uses implicit functions to express the spatial geometry field, such as signed distance functions (SDF) [37] and occupancy fields [38]. While implicit function-based methods [1, 39, 6] can accurately reconstruct the complex topology of clothed human body surfaces, they may generate non-anatomical shapes for out-of-distribution poses due to the lack of regularization.

To improve pose robustness, recent research [3, 2, 4, 5, 7] has utilized the prior knowledge to guide implicit function representation. These methods have shown promising results in enhancing the quality and accuracy of reconstruction geometry, particularly for challenging poses. However, these methods, like previous ones, still rely on 2D features extracted from CNN-based networks, even though some of them incorporate 3D features obtained from human body prior. In the reconstruction process, the feature obtained by 2D projection may result in incomplete reconstructions from other viewpoints and diminish overall reconstruction accuracy. Our method extracts global-correlated 3D-aware feature to efficiently represent the clothed human avatar.

**Transformers in Vision.** The transformer architecture, initially proposed by Vaswani et al. [40], has achieved immense success across domains like NLP, speech recognition, and multimodal applications. Inspired by this, many studies have attempted to adapt the transformer architecture to the field of computer vision. Among these explorations, the Vision Transformer (ViT) proposed by Dosovitskiy et al. [41] has shown impressive performance in 2D visual tasks. Meanwhile, transformer's ability to model global and long-range correlation is also suitable for 3D vision tasks. Therefore, we leverage a ViT-based 3D transformer with cross-plane attention to efficiently extract global-correlated 3D features for better human reconstruction.

**Generative 3D-aware Feature.** Recent studies [14] have proposed the tri-plane 3D feature representation method, which efficiently extracts features from objects in three orthogonal orientations. Tri-plane representation has been demonstrated high efficacy in generating 3D objects [42], particu-

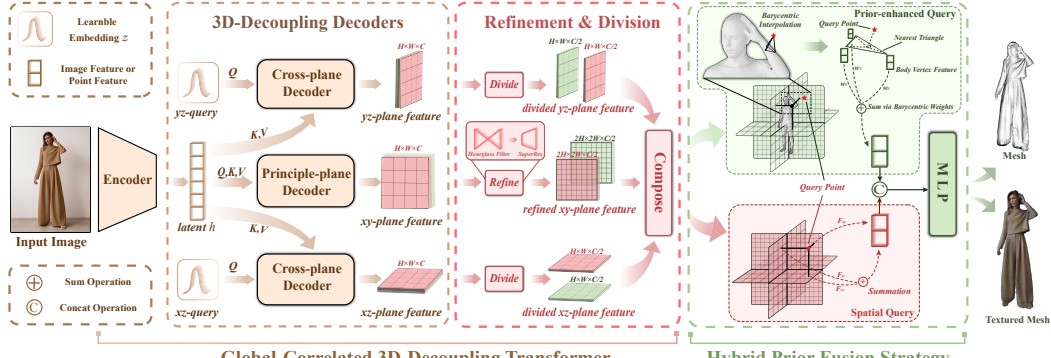

Figure 3: GTA Overview. GTA has two key modules: (1) the global-correlated 3D-decoupling transformer and (2) the hybrid prior fusion strategy. The former module extracts a latent $h$ from the input image and integrates it with learnable embeddings $z$ via 3D-decoupling decoders, producing disentangled tri-plane features. The latter module employs spatial (red) and prior-enhanced (green) queries to merge features from the two tri-planes, thus enabling geometry and texture reconstruction.

larly 3D human bodies [43, 44, 45]. Thus, the tri-plane representation method holds potential for application in human body reconstruction. Nevertheless, the challenge of establishing a reasonable relationship between the input monocular image and the three planes of features remains an unresolved problem. In our method, we introduce learnable embeddings to represent features of spatial planes that are not directly visible, and utilize cross-attention mechanisms to establish relationships between the input image and other planes.

## 3 Method

We introduce an implicit function-based framework for reconstructing 3D clothed human models from a single image (See Fig. 3). Our model employs a global-correlated 3D-decoupling transformer to disentangle tri-plane features and harnesses a hybrid prior fusion strategy for reconstructing the full 3D geometry and texture of the clothed avatar. In the following sections, we will discuss the preliminaries of **GTA** in Sec. 3.1, the global-correlated 3D-decoupling transformer in Sec. 3.2, and the hybrid prior fusion strategy in Sec. 3.3.

### 3.1 Preliminary

**SMPL.** The Skinned Multi-Person Linear (SMPL) model [18] is a widely-used parametric human body model. The SMPL model utilizes shape parameters $\boldsymbol{\beta} \in \mathbb{R}^{10}$ and pose parameters $\boldsymbol{\theta} \in \mathbb{R}^{3 \times K}$ to parameterize the deformation of the human body mesh $\mathcal{M}$:

$$\mathcal{M}(\boldsymbol{\beta}, \boldsymbol{\theta}) : \boldsymbol{\beta} \times \boldsymbol{\theta} \mapsto \mathbb{R}^{3 \times N} \tag{1}$$

where $K = 24$ joints and $N = 6890$ vertices. Shape parameter $\boldsymbol{\beta}$ describes the body's overall size and proportions and pose parameter $\boldsymbol{\theta}$ defines the positions and orientations of the joints relative to their default positions. SMPL enables effective representation and manipulation of human body shape and pose in various applications.

**Implicit Function.** Implicit function is a powerful tool for modeling complex geometries with neural networks. Our implicit function maps an input point to a scalar value that represents the spatial field including occupancy field and color field. The occupancy field takes a point in space as input and outputs a binary value indicating whether the point is inside or outside the human surface. Our reconstructed human surface can be represented as $\mathcal{S}_{\mathcal{IF}}$:

$$\mathcal{S}_{\mathcal{IF}} = \{\boldsymbol{x} \in \mathbb{R}^3 \mid \mathcal{IF}(\boldsymbol{x}) = (o, \boldsymbol{c})\} \tag{2}$$

where occupancy $o = 0.5$, color $\boldsymbol{c} \in \mathbb{R}^3$, and $\mathcal{IF}$ represents the implicit function.

## 3.2 Global-correlated 3D-decoupling Transformer

Directly extracting 3D information from a single 2D image is not feasible, this is largely due to the fact that 3D features include information from planes orthogonal with the image plane (also noted as $xy$-plane, principle-plane). To effectively decouple cross-plane features from a monocular image input, it is crucial to have additional guiding information. This is where the idea of learnable embeddings and the use of a cross-attention mechanism prove valuable. Inspired by this, we devise a novel transformer-based architecture including a global-correlated encoder and a 3D-decoupling decoder to disentangle 3D features from single input images.

**Global-correlated Encoder.** In our method, we employ a vision transformer to encode the input image and capture global correlations in the image, resulting in high-dimensional global-correlated image features. Our encoder module processes the input image by dividing it into non-overlapping $n \times n$ patches and subsequently mapping them to image features through transformer blocks. This procedure generates a latent $h$ for a image $I$.

**3D-decoupling Decoder.** To decode 3D tri-plane features from the encoder output, we propose to use two types of decoders: the principle-plane decoder and the cross-plane decoder. Our principle-plane decoder generates $xy$-plane features, which share the same plane as the input image. This decoder effectively reverses the encoding operation, leveraging a self-attention mechanism on the encoder output and converting the image features into a principal feature map $F_{xy} \in \mathbb{R}^{H \times W \times C}$ .

In order to generate plane features orthogonal with the principle plane while preserving global correlation with the principle plane, we employ the cross-plane decoder to decode $yz$ and $xz$ plane features from input image features. To guide the decoder in decoding features from different planes, we introduce a learnable embedding $z$ that supplies additional information for decoupling new planes. The learnable embedding $z$ is first processed through self-attention encoding. It is then used as a query in a multi-head cross-attention mechanism with the output image latent $h$ from the encoder stack. The image features are converted into keys and values for the cross-attention mechanism,

$$\mathbf{CrossAttn}(z, h) = \mathbf{Softmax}\left(\frac{(W^Q \mathbf{SelfAttn}(z))(W^K h)^T}{\sqrt{d}}\right)(W^V h) \tag{3}$$

where $W^Q$, $W^K$, and $W^V$ are learnable parameters and $d$ is the scaling coefficient. Following the original transformer architecture [40], our model employs residual connections [46] and layer normalization [47] after each sub-layer. The entire decoder consists of multiple identical layers, and we use two such decoders to produce feature maps $F_{yz} \in \mathbb{R}^{H \times W \times C}$ and $F_{xz} \in \mathbb{R}^{H \times W \times C}$.

**Principle-plane Refinement.** In accordance with the approach demonstrated in [6], higher-resolution feature maps play a crucial role in producing detailed geometry and sharper textures. Hence, we use both the original image and the principle-plane feature map, to produce a higher-resolution feature map. The original image is initially down-convoluted to match the principle-plane size and then concatenated along the channel dimension. Subsequently, they are fed into a streamlined Hourglass network and a super-resolution module for refinement. This process generates a higher-resolution feature map $F_{xy}^{refine} \in \mathbb{R}^{2H \times 2W \times C}$.

$$F_{xy}^{refine} = \mathbf{SuperRes}(\mathbf{Hourglass}(\mathbf{DownConv}(I) \copyright F_{xy})) \tag{4}$$

where $\copyright$ means concatenation operation. The resulting $xy$ plane (principle-plane) exhibits a higher resolution than the $yz$ and $xz$ planes and incorporates more information from the original image, leading to higher fidelity reconstruction.

**Tri-plane Division.** After obtaining each plane, we evenly divide the plane features along the channel dimension into two groups, creating two tri-planes. For one group, we perform a spatial query to acquire features for query points, while for the other group, we utilize a prior-enhanced query to integrate the human body prior. Please refer to the next section for a detailed introduction to our novel hybrid prior fusion strategy.

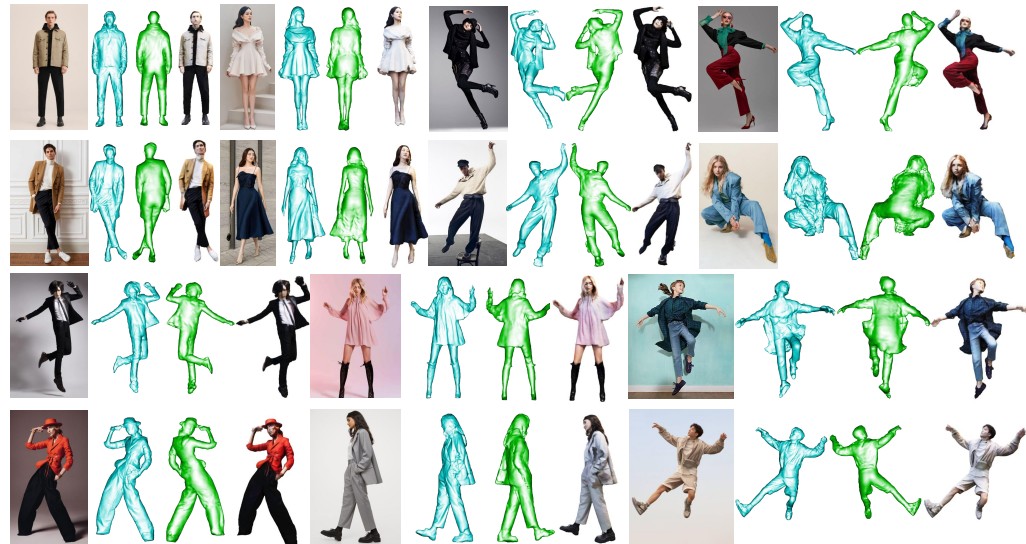

Figure 4: Qualitative 3D human reconstruction for real images showcasing diverse poses and clothing variations. For each example, we show the input image along with two views of the reconstructed geometry and front view of the reconstructed texture. Our approach is robust to challenging poses and loose clothing, and contains detailed geometry and texture. See SupMat. for more results.

### 3.3 Hybrid Prior Fusion Strategy

In previous works, two primary methods have been utilized for acquiring query point features, each with significant limitations as previously discussed in Sec. 1. To address this, we propose a hybrid prior fusion strategy that combines the strengths of both spatial query and prior-enhanced query.

**Spatial Query.** Following [14], we extend the pixel-aligned query into 3D space, denoted as spatial query. This method projects query points onto the $xy$, $yz$, and $xz$ planes of a tri-plane group, producing localized features that capture important details for reconstruction. We combine the $F_{yz}$ and $F_{xz}$ features by summation and concatenate the result with $F_{xy}^{refine}$ to generate the spatial query feature $F_{SQ}(\boldsymbol{x})$:

$$F^{SQ}(\boldsymbol{x}) = F_{xy}^{SQ}(\boldsymbol{x}) \copyright (F_{yz}^{SQ}(\boldsymbol{x}) + F_{xz}^{SQ}(\boldsymbol{x})) \tag{5}$$

where $F_{xy}^{SQ}(\boldsymbol{x})$, $F_{yz}^{SQ}(\boldsymbol{x})$, $F_{xz}^{SQ}(\boldsymbol{x})$ are extracted from $F_{xy}^{refine}$, $F_{yz}$, $F_{xz}$, and $\copyright$ is concatenation.

**Prior-enhanced Query.** For the other tri-plane, we project the human body prior [18, 19] mesh vertices onto the three planes similar to the spatial query above to obtain the feature $F^{PQ}(\boldsymbol{v})$, $\boldsymbol{v} \in \mathcal{M}$, where $\mathcal{M}$ is the body prior mesh. For each query point $\boldsymbol{x}$, we find the nearest triangular face $t_{\boldsymbol{x}} = [\boldsymbol{v}_0, \boldsymbol{v}_1, \boldsymbol{v}_2] \in \mathbb{R}^{3\times3}$ and use barycentric interpolation to integrate features for $\boldsymbol{x}$ (See Fig. 3), denoted as $F_{PQ}(\boldsymbol{x})$:

$$F^{PQ}(\boldsymbol{x}) = u F^{PQ}(\boldsymbol{v}_0) + v F^{PQ}(\boldsymbol{v}_1) + w F^{PQ}(\boldsymbol{v}_2) \tag{6}$$

where $[u, v, w]$ represents the barycentric coordinates of the query point $\boldsymbol{x}$ projected onto triangle $t_{\boldsymbol{x}}$.

**Hybrid Prior Fusion Strategy.** Spatial query projects the query points directly onto tri-plane features, providing detailed information but lacking prior knowledge. On the other hand, the prior-enhanced query merges body prior information but may causes an increased level of fuzziness. Therefore, we concatenate these two query features to capitalize on each method's strengths and compensate for their weaknesses. Furthermore, we also include the signed distance between query point and human prior mesh $\mathcal{SDF}_{Prior}(\boldsymbol{x})$ and pixel-aligned normal feature $F_{\mathcal{N}}(\boldsymbol{x})$ as input to the implicit function for predicting occupancy and color. Consequently, the reconstructed human surface $\mathcal{S}_{\mathcal{IF}}$ can be represented as:

Table 1: Quantitative comparison on geometry against other methods. *: obtained from [2, 4].

| Method | Training Data | CAPE-NFP [16] | | | CAPE-FP [16] | | | THuman2.0 [15] | | |
|---|---|---|---|---|---|---|---|---|---|---|
| | | Chamfer ↓ | P2S↓ | Normals↓ | Chamfer ↓ | P2S↓ | Normals↓ | Chamfer ↓ | P2S↓ | Normals↓ |
| PIFu [1] | THuman2.0 [15] | 2.458 | 2.117 | 0.094 | 1.786 | 1.639 | 0.071 | 1.586 | 1.530 | 0.088 |
| PIFu* [1] | Renderpeople [48] | 2.973 | 2.940 | 0.111 | 2.100 | 2.093 | 0.091 | - | - | - |
| PIFuHD* [6] | Renderpeople [48] | 3.767 | 3.591 | 0.123 | 2.302 | 2.335 | 0.090 | - | - | - |
| PaMIR [3] | THuman2.0 [15] | 1.603 | 1.429 | 0.068 | 1.502 | 1.291 | 0.064 | 1.276 | 1.247 | 0.080 |
| PaMIR* [3] | Renderpeople [48] | 1.413 | 1.321 | 0.063 | 1.225 | 1.206 | 0.055 | - | - | - |
| ICON [2] | THuman2.0 [15] | 1.096 | 1.085 | 0.046 | 0.969 | 0.987 | 0.041 | 1.249 | 1.368 | 0.076 |
| ICON* [2] | Renderpeople [48] | 1.070 | 1.013 | 0.059 | 1.202 | 1.170 | 0.055 | - | - | - |
| ECON [4] | THuman2.0 [15] | 0.942 | 0.933 | 0.035 | 0.904 | 0.894 | 0.033 | 2.120 | 1.807 | 0.074 |
| ECON* [4] | THuman2.0 [15] | 0.926 | 0.917 | 0.037 | - | - | - | - | - | - |
| **Ours** | THuman2.0 [15] | **0.911** | **0.917** | 0.042 | **0.763** | **0.763** | 0.035 | **0.814** | **0.862** | **0.055** |

$$\mathcal{S}_{\mathcal{IF}} = \{\boldsymbol{x} \in \mathbb{R}^3 \mid \mathcal{IF}(F^{SQ}(\boldsymbol{x}), F^{PQ}(\boldsymbol{x}), \mathcal{SDF}_{Prior}(\boldsymbol{x}), F_{\mathcal{N}}(\boldsymbol{x})) = (o, \boldsymbol{c})\} \tag{7}$$

where occupancy $o = 0.5$, color $\boldsymbol{c} \in \mathbb{R}^3$, and $\mathcal{IF}$ represents the implicit function.

**Training Objectives.** For each 3D scan, we consider two sets of points as training data, denoted as $G_o$ and $G_c$. $G_c$ is sampled uniformly with a slight perturbation along the normals of the mesh surface, whereas $G_o$ is sampled according to the same strategy as in PIFu [1], where points are sampled near the mesh surface and throughout the entire space.

For the points in $G_o$, we employ the following loss function:

$$\mathcal{L}_o = \frac{1}{|G_o|} \sum_{\boldsymbol{x} \in G_o} BCE(\hat{o}_{\boldsymbol{x}} - o_{\boldsymbol{x}}) \tag{8}$$

where $\hat{o}_{\boldsymbol{x}}$ denotes the model's predicted occupancy, while $o_{\boldsymbol{x}}$ signifies the ground truth occupancy. For the sampled points in $G_c$, we apply the following loss function:

$$\mathcal{L}_c = \frac{1}{|G_c|} \sum_{\boldsymbol{x} \in G_c} |\hat{\boldsymbol{c}}_{\boldsymbol{x}} - \boldsymbol{c}_{\boldsymbol{x}}| \tag{9}$$

where $\hat{\boldsymbol{c}}_{\boldsymbol{x}}$ represents the predicted color at location $\boldsymbol{x}$ by the model, while $\boldsymbol{c}_{\boldsymbol{x}}$ indicates the true color of the mesh at $\boldsymbol{x}$. The overall loss function is expressed by:

$$\mathcal{L}_{GTA} = \mathcal{L}_o + \mathcal{L}_c \tag{10}$$

## 4 Experiments

**Strategy for Point Sampling.** In the context of each training subject, our approach involves obtaining 2048 points for occupancy, denoted as $G_o$, and 2048 points for color, symbolized as $G_c$. The method for occupancy point sampling is aligned with the strategy illustrated in [1]. Color points are sampled uniformly, with a minor Gaussian disturbance, expressed as $\mathcal{N}(0, \sigma)$, wherein our experiment $\sigma$ is set at 0.1 cm. This disturbance occurs along the normals of the mesh surface. We obtain labels for the ground truth geometry, which specify whether a point is inside or outside the surface, through the application of Kaolin [49] to ascertain if a point lies within the ground truth mesh. The source of the ground truth color labels is the UV texture map of the 3D meshes.

**Model Structure.** To generate global-correlated latent features, we utilize a Vision Transformer (ViT) [41] model of depth 6, functioning as our global-correlated encoder, and generating an output of size $1024 \times 256$. Our 3D-decoupling decoder incorporates both cross-plane and principal-plane decoders, each with a depth of three. The cross-plane decoder is initialized with learnable embeddings that experience a Gaussian perturbation to align cohesively with the encoder's output shape. The configuration of the cross-plane decoder corresponds with the structure described in [40], while the principal-plane decoder emulates the global-correlated encoder. Each decoder outputs a feature map $F \in \mathbb{R}^{128 \times 128 \times 64}$. During refinement, a 2-stack hourglass and a transpose convolution module are integrated to generate a higher resolution principal-plane feature map $F_{xy}^{refine} \in \mathbb{R}^{128 \times 128 \times 64}$. Following the feature acquisition through our hybrid prior fusion method, two identical Multilayer

Perceptrons (MLPs) are employed for separate predictions of occupancy and color, each with layer sizes of [512, 1024, 512, 256, 128, 1]. In the inference phase, we utilize Rembg [50] for background subtraction in in-the-wild images. The Marching Cubes algorithm [51] is employed for generating 3D meshes, while off-the-shelf models from ICON [2] are leveraged for the production of normal maps. This normal map is further processed through a 2-stack hourglass to achieve a size of $128 \times 128 \times 6$. Besides the front/back normal maps are also used as input into the encoder with the image. The model, implemented in PyTorch Lightning [52], is trained for 10 epochs with a learning rate of 1e-4 and a batch size of 4, over a span of 2 days on a single NVIDIA GeForce RTX 3090 GPU.

**Datasets.** Our model was trained on the THuman2.0 [15], featuring 526 high-quality human scans, with 505 designated for training and 21 for evaluation. Testing was primarily conducted on the CAPE [16] and THuman2.0, with the former divided into "CAPE-FP" and "CAPE-NFP" subsets to examine model generalization on different pose types. Further dataset and implementation details are available in SupMat.

**Metrics.** We employ Chamfer and point-to-surface (P2S) distances, capturing significant geometric errors, to evaluate the accuracy of the

Table 2: Normal Evaluation of Different Views on THuman2.0 [15]. These views are obtained by positioning a virtual camera at the front, left, back, right, above, and below the reconstructed human.

| Method | Normals of Different Views | | | | | | Average |
| | Front | Left | Back | Right | Above | Below | |
| --- | --- | --- | --- | --- | --- | --- | --- |
| PIFu [1] | 0.053 | 0.143 | 0.046 | 0.109 | 0.067 | 0.066 | 0.081 |
| PaMIR [3] | 0.051 | 0.124 | 0.068 | 0.077 | 0.051 | 0.054 | 0.071 |
| ICON [2] | 0.074 | _0.091_ | 0.0647 | _0.076_ | _0.044_ | 0.044 | 0.066 |
| ECON [4] | **0.043** | 0.123 | _0.045_ | 0.083 | 0.050 | _0.041_ | _0.064_ |
| **Ours** | _0.048_ | **0.069** | **0.044** | **0.061** | **0.035** | **0.040** | **0.050** |

reconstructed meshes. We assess the quality of local details and the efficacy of 3D features via the L2 error between normal images of reconstructed and ground-truth meshes from six views. Finally, the quality of texture prediction is measured using Peak Signal-to-Noise Ratio (PSNR), comparing images rendered from both reconstructed and ground-truth surfaces across different views.

## 4.1 Evaluation

**Evaluation of Geometry.** We compare our **GTA** model with body-agnostic methods like PIFu [1], PIFuHD [6], and body-aware methods such as PaMIR [3], ICON [2], and ECON [4]. Our evaluation is thorough, involving training and testing these models ourselves and incorporating testing results from [4, 2] for a comprehensive comparison. As depicted in Tab. 1, **GTA** excels in terms of Chamfer and P2S distances on images with out-of-distribution (OOD) poses and diverse clothing. Notably, **GTA** is the first to reduce the Chamfer distance to less than 0.8 cm on CAPE-FP. On par with ECON for normals on CAPE, **GTA** sets a new state-of-the-art on THuman2.0. Fig. 8a visually underlines our model's superior performance on the THuman2.0 benchmarks.

**Quantitative Evaluation of Side-face Reconstruction.** In our novel quantitative evaluation of side-face reconstruction, we spotlight the advantages of 3D features in crafting plausible side-faces and accurate thickness. Using six virtual camera angles, we render normal images of the reconstructed human and compute the normal difference for each face on the THuman2.0 [15] dataset. As shown in Tab. 2, **GTA** surpasses other models in 5 out of 6 views, matching ECON only in the front view, thereby underscoring our model's prowess in capturing inherent 3D structures within an image. For visual results, refer to Fig. 2 and SupMat.

**Evaluation of Texture.** In evaluating texture reconstruction, we compare **GTA** with color-predicting models like PIFu [1], ARCH [8], ARCH++ [9], PHORHUM [5], and S3F [7]. By rendering textured meshes from multiple angles and calculating the PSNR with respect to ground truth images, we find **GTA** outperforms other models on THuman2.0. As Fig. 5 demonstrates, **GTA** exceeds the state-of-the-art S3F [7] by 22% in PSNR. Notably, our model provides superior textures on the front side and accurately predicts invisible regions, as seen in Fig. 8b, highlighting the effectiveness of 3D features.

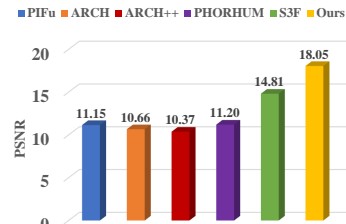

Figure 5: Quantitative Evaluation of Texture on THuman2.0 [15].

## 4.2 Ablation Study

**Ablation Details.** To ensure a fair and unbiased experimental setup, we employed an approach wherein the implicit functions of each ablation experiment were augmented with front and back

Table 3: Ablation study of several of our designs on CAPE [16] dataset.

(a) Different networks.

| Method | Chamfer↓ | P2S↓ | Normals↓ |
|---|---|---|---|
| use convolution | 0.991 | 0.968 | 0.055 |
| w/o cross-atten | 0.937 | 0.922 | _0.051_ |
| w/o refine | _0.890_ | _0.882_ | 0.053 |
| **Ours** | **0.861** | **0.866** | **0.045** |

(b) 2D features vs. 3D features.

| Method | Chamfer↓ | P2S↓ | Normals↓ |
|---|---|---|---|
| 2D+SQ | 1.054 | 1.052 | 0.054 |
| 2D+PQ | 1.133 | 1.116 | 0.053 |
| 2D+hybrid | _1.008_ | _0.967_ | _0.051_ |
| **Ours** | **0.861** | **0.866** | **0.045** |

(c) Different query methods.

| Method | Chamfer↓ | P2S↓ | Normals↓ |
|---|---|---|---|
| 3D+SQ | _0.987_ | _0.965_ | 0.049 |
| 3D+PQ | 1.059 | 0.987 | _0.048_ |
| **Ours** | **0.861** | **0.866** | **0.045** |

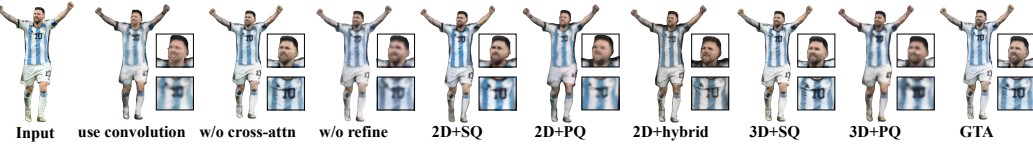

Input    use convolution    w/o cross-attn    w/o refine    2D+SQ    2D+PQ    2D+hybrid    3D+SQ    3D+PQ    GTA

Figure 6: Texture change of different ablation settings.

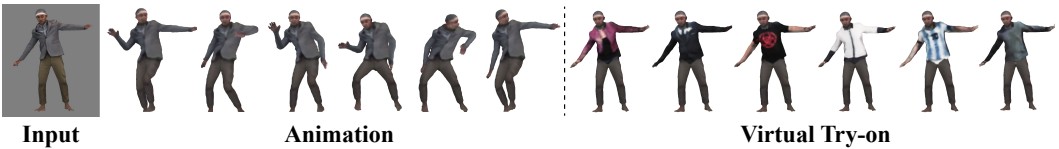

**Input**        **Animation**        **Virtual Try-on**

Figure 7: Application of our model in animation and virtual try-on.

normal features and signed distances from the human prior body. For the ablation experiments pertaining to different network architectures, we replaced the corresponding components of our network structure with three identical UNet [53] and Vision Transformer [41] decoders solely based on self-attention, respectively. Moreover, for the ablations on 2D features, we generated a 2D feature map of size $256 \times 256$ with UNet. Additionally, for the ablations on the hybrid prior fusion strategy, we employed a single tri-plane for the spacial query and the prior-enhanced query, respectively.

**A. Different Networks for 3D Feature Decoupling.** We evaluate alternative architectures by modifying our global-correlated encoder and 3D-decoupling decoder, confirming the strength of our proposed transformer. We experiment with UNet [53], also used in [7], as a convolution filter representative, and employ three separate UNets to compose the tri-plane. We also test a transformer encoder-decoder with only self-attention and without the refinement module or additional learnable embeddings. Results (see Tab. 3 and Fig. 6) suggest that a purely convolution-based network struggles to decouple 3D features due to limited correlation and receptive field constraints. While the refinement module shows minor geometric improvements, it is crucial for texture reconstruction. Additional learnable embeddings and cross-attention blocks notably enhance geometry results.

**B. 2D Features vs. 3D Features.** We analyze the effectiveness of 3D features by conducting an ablation study using solely 2D feature maps for reconstruction. Utilizing UNet [53] to generate high-dimensional 2D features, we compare spatial query (SQ), prior-enhanced query (PQ), and our hybrid prior fusion strategy. Results highlight the inferiority of 2D features in producing accurate reconstructions (Tab. 3), emphasizing the importance of 3D features.

**C. Hybrid Prior Fusion Strategy vs. Others.** We evaluate our hybrid prior fusion strategy against individual use of spatial query (SQ) and prior-enhanced query (PQ). Results show that spatial query surpasses prior-enhanced query in both geometry and texture quality due to its provision of localized features for detailed reconstruction. However, combining both methods optimizes geometry performance while preserving texture quality, demonstrating the effectiveness of our hybrid strategy.

## 4.3 Applications

**Reconstruction of Images in-the-wild.** The **GTA** model demonstrates significant prowess in reconstructing 3D human meshes from unconstrained, real-world images (refer to Fig. 4 and SupMat.), addressing the complexities posed by varied poses and clothing styles. This capability of recon-

structing high-fidelity 3D models from in-the-wild images paves the way for extensive applications, notably in virtual and augmented reality.

**Animation and Virtual Try-On.** We present a robust approach for generating novel poses of 3D clothed human meshes, catering to applications in animation and virtual try-on (See Fig. 7). We extend the S3F [7] model by employing estimated body shape and pose parameters to derive tri-plane features, facilitating realistic deformations. For a single-image clothed human reconstruction, our method excels by only needing the target pose, thus overcoming the limitations of previous deep learning methods. Additionally, our model supports virtual try-on by enabling feature replacement across body parts of different parametric bodies. By selectively interchanging these features, we can simulate changes in clothing on the target image. Our method, therefore, provides a versatile solution for both animation and virtual try-on applications, merging the strengths of previous methods while alleviating their weaknesses. More technical details and results are available in SupMat.

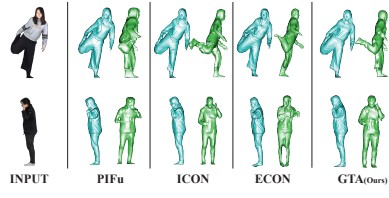

(a) Qualitative comparison of geometry.

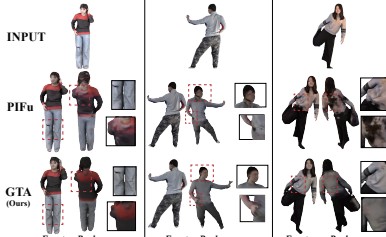

(b) Qualitative comparison of texture.

Figure 8: Comparison of geometry and texture on THuman2.0 [15] benchmark.

## 5   Conclusion

In conclusion, we present the **GTA** model, a cutting-edge approach for reconstructing 3D clothed human from single images. Our global-correlated 3D-decoupling transformer effectively extracts latent representations from input images and integrates them with learnable embeddings through 3D-decoupling decoders, generating disentangled tri-plane features. Moreover, our hybrid prior fusion strategy integrates the benefits of spatial query's localization capabilities with the prior-enhanced query's ability to incorporate knowledge of the human body prior, ultimately leading to improved geometry and texture reconstruction performance. We demonstrate that our proposed model outperforms state-of-the-art methods in geometric and texture reconstruction, exhibiting resilience against challenging poses and loose clothing, enabling a wide range of applications.

**Acknowledgements.** This work was supported in part by the Fundamental Research Funds for the Central Universities (No. 226-2023-00048) and the Natural Science Foundation of Zhejiang Province (DT23F020008).

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
