# Supplementary Material of Global-correlated 3D-decoupling Transformer for Clothed Avatar Reconstruction

**Zechuan Zhang**[1], **Li Sun**[1], **Zongxin Yang**[1], **Ling Chen**[2], **Yi Yang**[1†]

[1] ReLER, CCAI, Zhejiang University
[2] AAII, University of Technology Sydney

## A    Implementation Details

**Estimation of Prior.** For the initial estimation of the human body, we resort to the utilization of either the SMPL-X model, as outlined in PIXIE [1], or the SMPL model, as delineated in PyMAF [2]. A variety of motion datasets like AIST++ [3] adopt SMPL parameters for animation representation, and hence we have used SMPL as the prior in the results depicted in Fig. 5.

During the training phase, the SMPL-X data tailored for THuman2.0 [4] is used as the human prior. For our geometry testing with open-source models, as delineated in Tab. **??**, we utilized the ground-truth SMPL/SMPL-X, consistent with the methodologies employed by ICON [5] and ECON [6]. In the texture performance evaluations with non-public models, such as S3F [7] in Fig. **??**, we abstained from using ground-truth SMPL and instead leveraged PyMAF [2] to derive the SMPL prior. While during the inference stage, we employ the readily available models [1, 2] to predict parameters for the human prior. In order to bolster the precision of the reconstructed outcomes, we deploy the refinement technique from [5], which allows us to optimize the SMPL/SMPL-X parameters for a reduced silhouette loss.

**Prior-guided Tri-plane Deformation.** Previous methods for generating novel poses of 3D clothed human meshes can be grouped into two categories. The first binds the 3D mesh with a human body prior, using algorithms like KNN [8], Surface Field [9], and MVC [10]. However, this approach has limited robustness and may cause distorted deformations in self-intersecting parts of the model. The second category employs deep learning to predict blend weights for each mesh point, enabling more realistic movements. These methods [11, 12, 13, 14] often require training on multiple 3D meshes of the same person in different poses, posing challenges for single-image clothed human reconstruction. Motivated by the S3F [7] approach, we employ the estimated body shape and pose parameters, denoted as $\beta$ and $\theta$, to obtain tri-plane features and transfer the features of each vertex in the prior model $\mathcal{M}(\beta, \theta)$ to the corresponding vertex in the posed prior $\mathcal{M}'(\beta, \theta')$, where $\theta'$ represents the target pose. Subsequently, we use barycentric interpolation to obtain the feature and feed them into the MLP to predict the 3D mesh and texture in the new pose.

**Virtual Try-on.** In our model, we can partition 3D features based on body parts (e.g., left arm, right leg) once integrated into a parametric human body. Given a reference and a target image, we feed them into our model to obtain two parametric bodies with vertex-specific features. We can then replace features on the target body's selected part (where clothes change is desired) with the corresponding features from the reference body. This replacement allows the target image's query point to acquire reference image features, enabling the implicit function to output colors from the reference image on the replaced parts.

---

† : the corresponding author.

37th Conference on Neural Information Processing Systems (NeurIPS 2023).

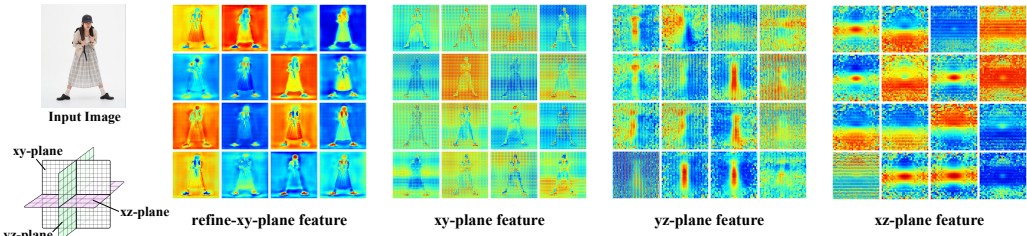

Figure 2: Tri-plane feature map visualization. The graphical display highlights the first 16 channels of our decoupled tri-plane features. These features effectively capture the human body's silhouette, demonstrating effectiveness of our approach.

# B   Experiments Analysis

We further analyze our experimental results and provide corresponding theoretical explanations.

Regarding quantitative evaluation metrics for geometry reconstruction, we observed that our approach achieved significant improvement in terms of Chamfer Distance and Point-to-Surface metrics, especially for in-distribution poses. The predominant factor of these advancements lies in our consideration of features of the orthogonal planes during reconstruction. Firstly, both of these metrics are employed to assess the large geometric differences. As exemplified by the results presented in Fig. 3a, ECON tends to exhibit larger errors in the orthogonal planes compared to GTA. These orthogonal plane errors manifest in three-dimensional space as overall surface shifts, significantly influencing chamfer distance and P2S metrics. Moreover, in terms of normal consistency, as ECON [6] explicitly integrates both front and back normal maps and performs explicit stitching for better hands and face details, our method yielded slightly lower results than ECON in the case of the CAPE [15] dataset. Conversely, in more intricate datasets like THuman2.0 [4], characterized by complexity and substantial occlusions, the disparities in the normal maps are pronounced, resulting in GTA exhibiting better overall Normals. In our further assessment of normal consistency, we observed that ECON [6] surpassed our method in terms of front view, which aligned with our analysis. For other views, our model produced rendered normal maps that more closely approximated the ground truth due to the utilization of our 3D features.

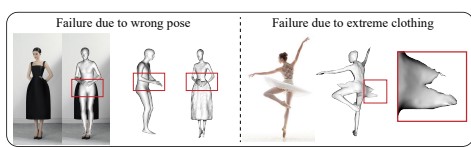

Figure 1: Failure cases. Inaccurate results due to incorrect human prior estimation (left) and failure cases due to extreme clothing (right).

The qualitative results of texture exhibit our method's superior performance in predicting textures of invisible regions. We attribute this success to our extraction of decoupled 3D features. Additionally, we conducted comprehensive ablation experiments, which provided evidence for the effectiveness of our network architecture, decoupled 3D features, and hybrid prior fusion strategy. Specifically, the results of the ablation experiments demonstrated that our network architecture, tri-plane features, and hybrid prior fusion strategy played indispensable roles in geometry reconstruction. Besides, the reconstruction of texture was primarily influenced by the principle plane refinement and spatial query.

# C   Limitations

Our method exhibits robust performance in general, though it does face challenges under certain circumstances, as depicted in Fig. 1. The model relies on pre-existing models [1, 2] to deduce a SMPL [16] or SMPL-X [17] from a single RGB image, thus errors in estimations can consequently affect our reconstruction. Additionally, our model faces difficulties with extremely loose clothing that considerably deviates from the human body's contours. To address these issues, future research could explore the integration of more explicit information during the reconstruction phase.

Table 1: Comparison of parameters and inference time.

(a) Parameter number comparison.

| Method | use convolution | ours |
|---|---|---|
| Total params | 93,135,264 | 36,977,056 |

(b) Inference time of different SOTA models.

| Method | PIFu | Pamir | ICON | ECON | GTA |
|---|---|---|---|---|---|
| Inference time (s) | 0.37 | 0.44 | 0.37 | 15.07 | 0.55 |

## D  Broader Impacts

Our model's ability to reconstruct 3D realistic avatars from single input images raises potential negative impacts, including privacy violations, "deep fakes" generation, and intellectual property infringement. To address these concerns, ethical guidelines and legal frameworks must be established, requiring collaboration among researchers, developers, and policymakers to ensure responsible applications of this technology.

## E  Additional Results

To further demonstrate the effectiveness of GTA, we compared it with the state-of-the-art methods in terms of time and space occupancy. In Tab. 1a, it's evident that our model outperforms the UNet-based approach with fewer parameters, as indicated by better reconstruction results. This highlights the advantages of our transformer-based design in extracting 3D features. Tab. 1b displays the comparable time efficiency of our implicit function-based model with PIFu, Pamir, and ICON. In contrast, ECON, relying on explicit methods, demands more time due to d-BiNI and Poisson inefficiencies. CAPE-NFP dataset with $256^3$ resolution is used for testing, ground truth SMPL/SMPL-X provided.

In order to assess the effectiveness of the decoupled tri-plane features, we conducted a visual analysis of the first 16 channels of these features in Fig. 2. The visualizations demonstrated that our refined principle plane features exhibited higher resolution and better visual results. While the features of the orthogonal planes appeared blurred, they still captured the overall outline of the human body. This observation serves as empirical evidence of the effectiveness of our approach, which incorporates 3D-decoupling decoders and principle plane refinement.

Fig. 3 compares the qualitative reconstruction results for geometry between the GTA and SOTA methods and showcases the impact of marching cubes resolution on GTA. In Fig. 3a, we provide rendered normal maps under the setting of quantitative testing (resolution of 256). The reconstruction results are rendered as normal maps from six different viewing angles. The results demonstrate that our approach is comparable to the SOTA methods for front-face reconstruction and outperforms them for side-face reconstruction. These qualitative results are consistent with the quantitative experiments presented in the main text and our analysis in the last section. In Fig. 3b, we illustrate various resolutions' effects on GTA's reconstruction results, elucidating the causes of detail loss.

Next, we build upon the results elucidated in the main paper by furnishing additional qualitative outcomes for various tasks. We present an extended range of monocular 3D clothed human reconstruction results in Fig. 4, demonstrating a wide assortment of input conditions in terms of poses, backgrounds, viewpoints, and clothing. Further, we showcase supplementary results of 3D human animation in Fig. 5 and 3D virtual try-on in Fig. 6 using images from Pinterest, SHHQ [18], and Deep Fashion [19]. While the animation and virtual try-on results offer a compelling demonstration of our model's capability, it's important to acknowledge that the resolution of these results does not quite match that of the reconstructed outcomes. This minor discrepancy arises from our approach that relies exclusively on the prior-enhanced query for these two applications. Although the current version of the spatial query cannot ensure smooth point deformation for these applications, which may cause minor artifacts, this represents a valuable avenue for potential improvement in future iterations. Nevertheless, the results of these two applications remain competitive when compared with other state-of-the-art methods.

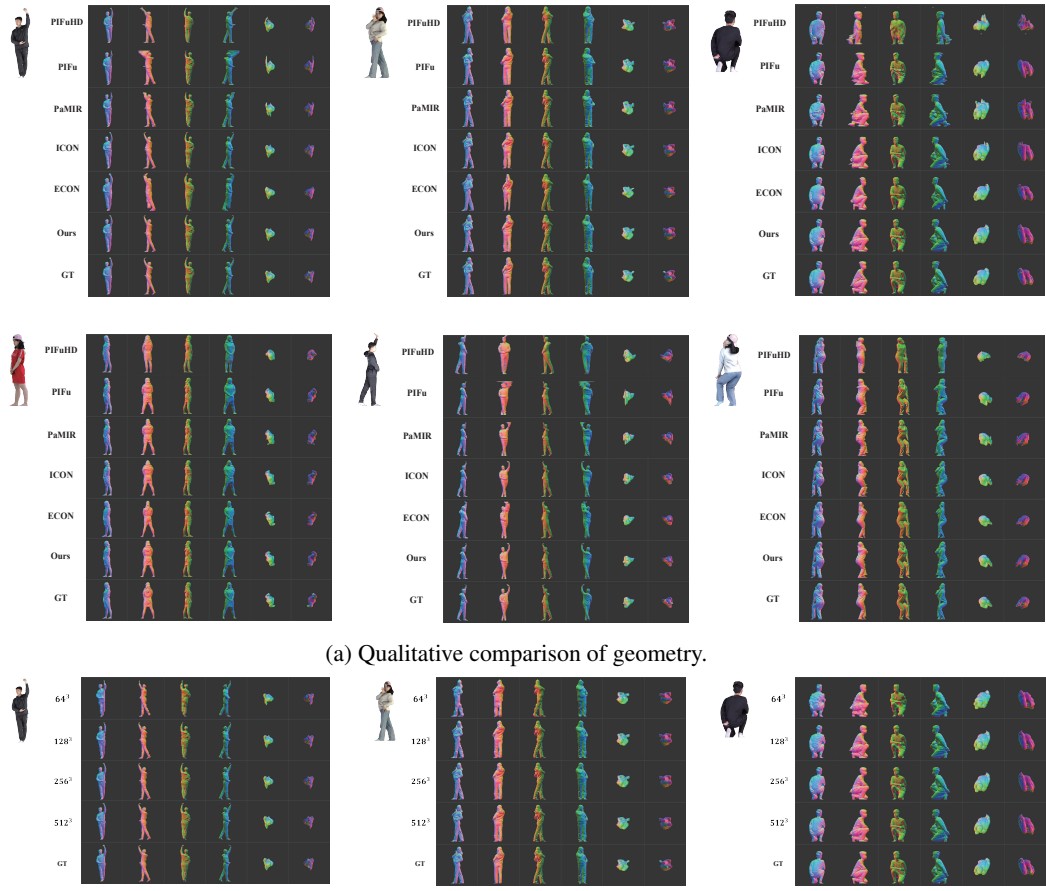

(a) Qualitative comparison of geometry.

(b) Qualitative comparison of different marching cube resolution.

Figure 3: Comparison of geometry on THuman2.0 benchmark. Please zoom in for details.

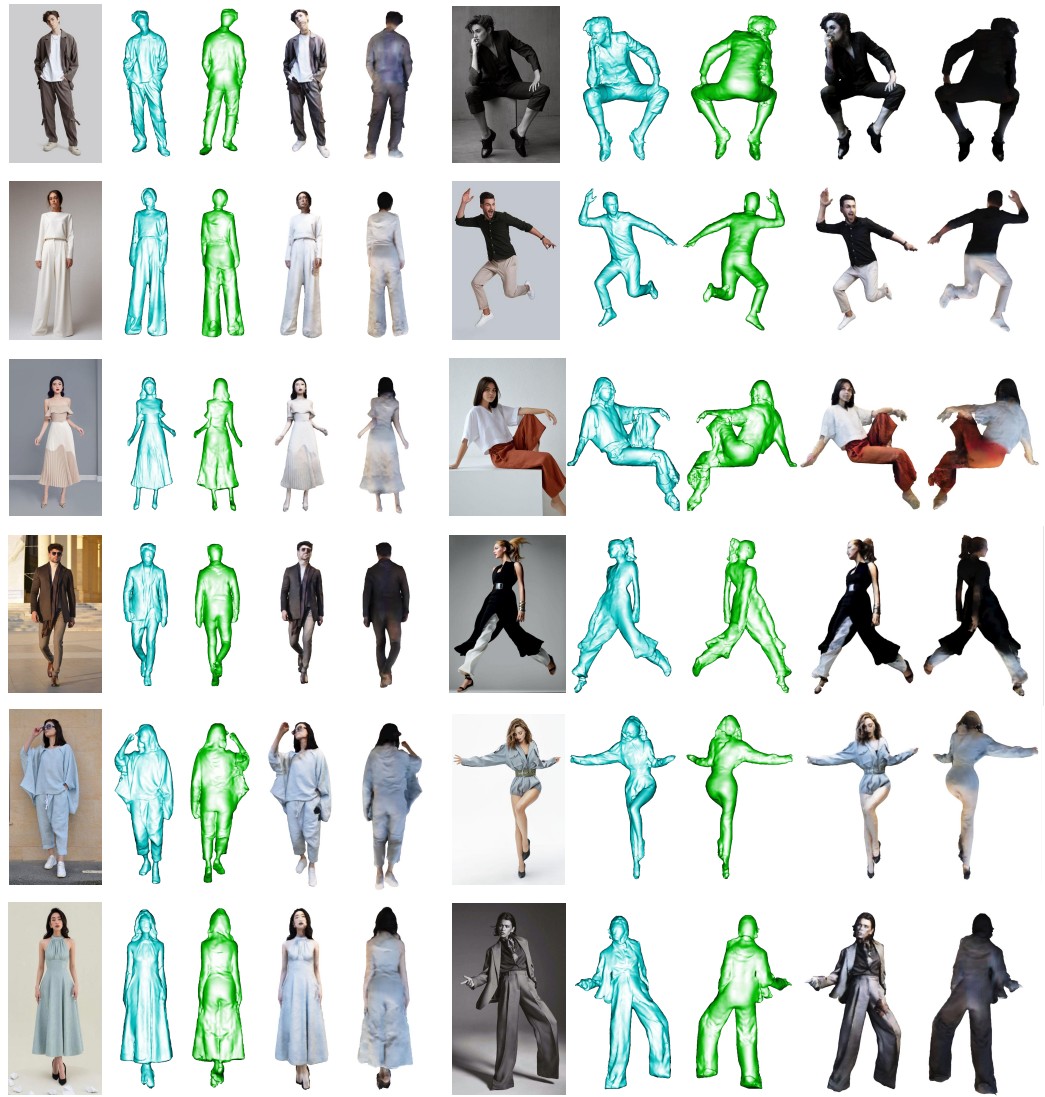

Figure 4: Qualitative 3D human reconstruction results on in-the-wild images with geometry and texture. For each example, we show the input image along with two views (front and back) of the reconstructed geometry and two views (front and back) of the reconstructed texture. Our approach is robust to pose variations, generalizes well to loose clothing, and contains detailed geometry and texture. Please zoom in for details.

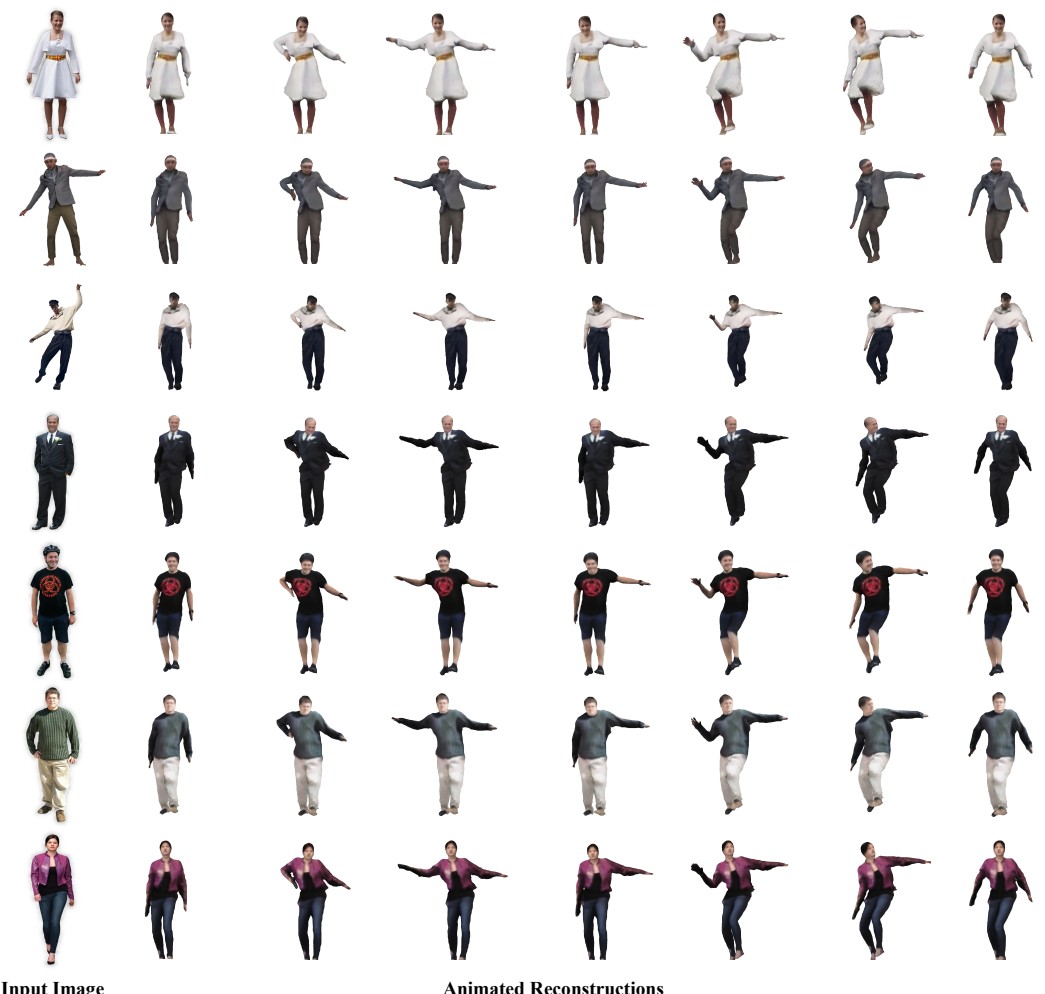

**Input Image**                    **Animated Reconstructions**

Figure 5: Qualitative results on animation of 3D reconstructions.

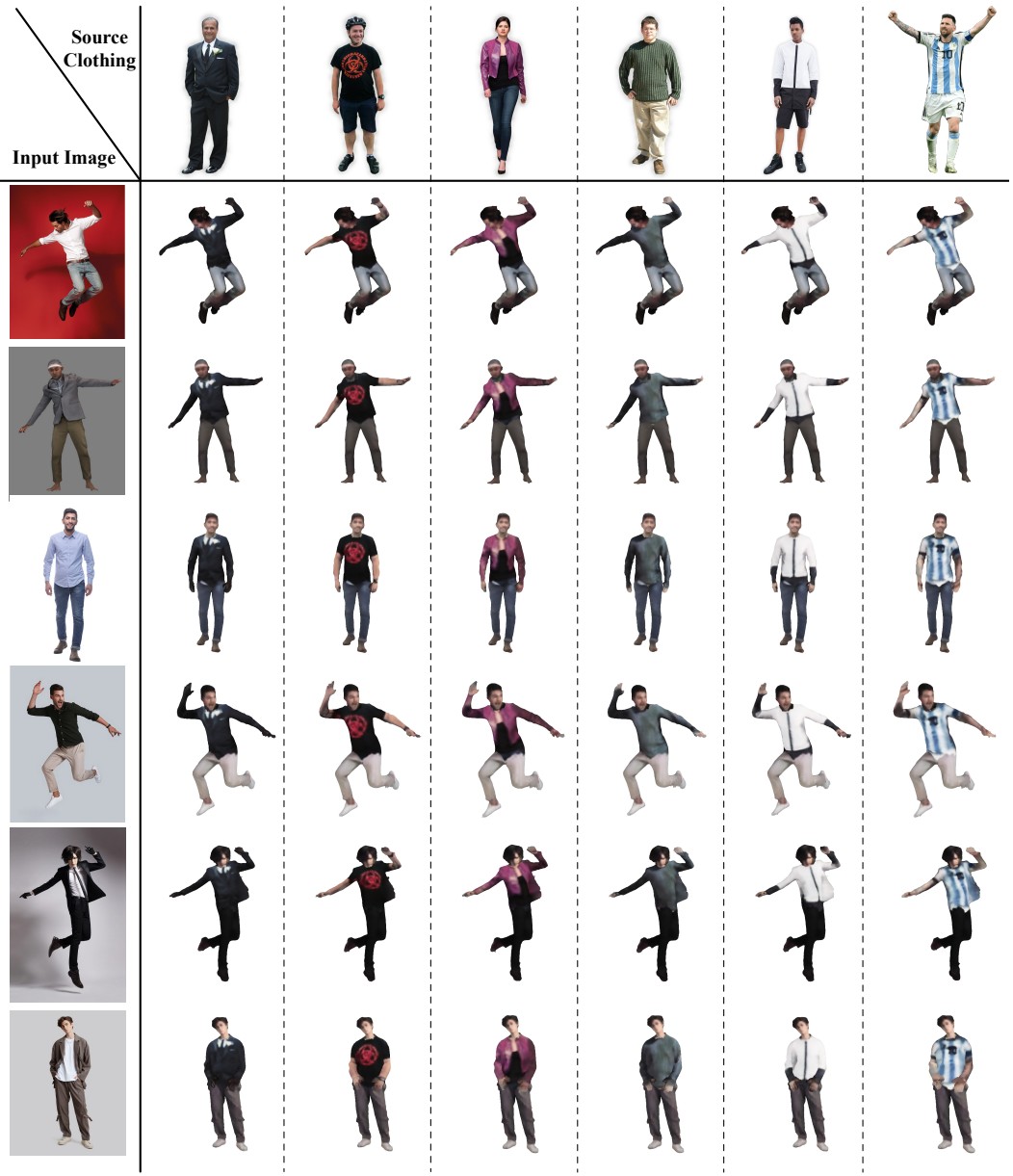

Figure 6: Qualitative results of 3D virtual try-on. We broaden the scope of the experiment discussed in the main paper by demonstrating an instance of upper-body clothing try-on and present both the input images (left) and source clothing (upper row). The 3D reconstructions generated exhibit high realism and consistency across all samples.