# OpenReview forum: "Global-correlated 3D-decoupling Transformer for Clothed Avatar Reconstruction"
_NeurIPS.cc/2023/Conference — NeurIPS 2023 poster_

### Official Review · Reviewer_6dkA · 2023-07-02

**Soundness:** 2 fair
**Presentation:** 3 good
**Contribution:** 2 fair
**Rating:** 5
**Confidence:** 5

**Summary:**

This paper aims at reconstructing 3D clothed human models from single images. Current methods rely heavily on 2D image features extracted from the input image, while ignoring information lying in the planes orthogonal to the input image plane. To address this limitation, this paper proposes a new method that utilizes attention mechanisms to construct tri-plane features in order to capture more 3D information. In addition, the authors also introduce a new feature query strategy that combines spatial query and manifold query. Results show that the proposed method outperforms existing baselines, showing high robustness to challenging poses and loose clothing.

**Strengths:**

Current methods typically use 2D convolutional networks to extract pixel-aligned features on the xy-plane, and rely on the networks themselves to infer 3D information along z-axis. In contrast, this paper explicitly considers the other two planes, i.e., the yz-plane and the xz-plane, and extracts features on these two planes as well. I think the insight behind the proposed method is valuable and can inspire future research.

**Weaknesses:**

1. My biggest concern is the inconsistent results. In Figure 1, the authors demonstrate a nice reconstruction result where geometric details are fairly recovered. However, in Figure 3 of the Supp.Mat., the reconstruction results of the proposed method are really coarse and many geometric details are missing. In addition, this figure shows that the proposed method performs significantly worse than ECON in terms of the front-view normal quality, although they should perform similarly according to Table 2 in the main paper.

2. Although the proposed method outperforms SOTA quantitatively, it comes at a cost of a much heavier network architecture. According to the method description, the proposed method uses multiple networks, including a vision transformer, several attention-based networks, an Hourglass network, an MLP as well as learnable embeddings. Compared to the networks used in PIFu/PaMIR/ICON/ECON, these networks are more complex and much heavier, containing a larger number of learnable parameters. Therefore, the performance gains may be due to "bigger" networks. Unfortunately, I do not see any discussion of method complexity in the paper.


3. Also, large networks may be prone to overfitting. However, it seems that the authors can train such large networks using a relatively small amount of data, i.e., only about 500 scans. It is not clear how the authors achieved this.

4. I think Figure 4 shoud show texture from more view points. The current version only shows the prediction results in the front view, which can be directly queried from the input image and cannot prove the texture estimation performance of the proposed method.

5. The learnable embeddings do not make sense to me. If I understand correctly, the learnable embeddings remain fixed after network training and are the same for different input images. However, for different images, the side view varies dramatically. Therefore, I don't think there exist unified embeddings that fit all input scenarios. I would like to see the visualization of learnable embeddings if possible.

6. Although I agree that solely relying on xy-plane is insufficient to reconstruct 3D models, I am suspicious whether the proposed method really solves this issue or not. According to Figure 2 in the Supp.Mat., the yz-plane and xz-plane feature maps do not contain much valid and distinguishable information compared to xy-plane. In other words, I guess that the learned network still relies heavily on xy-plane to reconstruct 3D models, and the robustness to challenging poses is mainly resulted from the introduction of prior-enhanced queries.

**Questions:**

1. Since the principle plane is refined after 3D-decoupling decoding, why not refine other planes (i.e., yz and xz planes) as well?

2. The authors keep emphasizing "global correlation" in the paper, but I cannot get the exact meaning of this term after reading the paper. Could the authors provide more detailed explanation or some examples to help reader understand what "global correlation" means?

**Limitations:**

The authors addressed the limitations and potential social impact in the supplemental document.

---

> ### Author Rebuttal · Authors · 2023-08-09
>
> We deeply appreciate your recognition of the valuable insight behind our method and its potential to inspire future research.
> We will address your inquiries and concerns point by point in the following responses.
>
>
> - **Answer to weakness 1**: Thank you for your keen observation. Because similar comments have been raised by other reviewers, we consolidate our response in the global author response "Answer for Unsatisfactory results in Figure 3 of SupMat".
>
> - **Answer to weakness 2**: Our results suggest that the performance improvement primarily arises from the intrinsic properties of the global-correlated 3D-decoupling transformer and the intricately crafted framework, rather than solely the parameter scale：
>   1. **Our transformer-based feature extractor possesses fewer parameters than the UNet-based counterpart but performs better.**
>   2. **Our model's time efficiency is comparable to other state-of-the-art (SOTA) methods that utilize implicit functions, while significantly outperforming ECON.**
>   3. **Please refer to global response for details about these comparisons.** In our forthcoming revisions, we will provide a clearer exposition of this issue in the supplementary material.
>
> - **Answer to weakness 3**:
>   1. **The training dataset we utilized, derived from THuman2.0, is substantial, which helps mitigate the risk of overfitting.** While it is true that we only selected about 500 scans from THuman2.0 for training, each scan was rendered in 36 views under various environmental lighting conditions. Consequently, the total number of data points used during training reached 18,180 (505 × 36), which is a considerably large dataset.
>   2. **Using a smaller dataset is also feasible in this field, but it can lead to longer training times.** For example, the S3F model [7], trained for five days on eight V100 GPUs with a batch size of 8, is larger than our model. Notably, our model, trained in two days on an NVIDIA GeForce RTX 3090 GPU with a batch size of 4, achieved competitive results. **S3F achieved SOTA performance using only 245 Renderpeople scans**, rendered five times each, as mentioned in [7] section 4.1 (Table 2, "Only synthetic data").
>   3. **The current state of this field lacks comprehensive research on the impact of training data size and the potential for overfitting.** We acknowledge and align with the reviewer's attention to this aspect. In the future, we intend to conduct further exploratory experiments to delve into these aspects more extensively.
>
>
>
> - **Answer to weakness 4**:
>   1. **We provide texture results from back side view and various view points in Figure 8 of the main text, Figure 4 in the supplementary material, and the SupMat's accompanying video.**
>   2. **We have meticulously computed PSNR for the outcomes rendered from multiple viewpoints.**
>   3. We greatly appreciate your suggestions and we are actively considering adjustments to the layout to incorporate texture results from different angles within the paper.
>
> - **Answer to weakness 5**:
>   1. **Despite the learnable embedding remaining fixed as query, the input image still serves as both the value and key inputs to the decoder.** This mechanism continues to decode features associated with different planes that maintain correlations with the image, as opposed to features unrelated to the image (refer to section 3.2 lines 166-172).
>   2. **The renowned object detection model DETR [a] also employs fixed learnable embeddings as queries in its decoder to detect objects in various input images.** This further illustrates the applicability of fixed learnable embeddings across different input images.
>   3. **In our ablation experiments, the reconstruction quality noticeably deteriorates when the learnable embedding is not employed** (as referenced in Section 4.2, lines 275-277, and "w/o cross-atten" in Table 3(a)). This compellingly underscores the significant role played by the incorporation of learnable embeddings.
>
>
>
> - **Answer to weakness 6**:
>   1. The features from the principal plane are crucial for reconstruction, yet previous methods have not considered incorporating features from the other two planes.
>   2. We address the challenges posed by heavy reliance on pixel-aligned features by introducing a novel tri-plane feature-based reconstruction approach. This concept, endorsed by reviewers Lf3d, rJsL, and 7DFs, is well-motivated, innovative, and efficacious.
>   3. The ablation study demonstrates the superiority of utilizing triplane features over relying solely on xy-plane features (2D+hybrid), as evidenced in Table 3(b).
>
> - **Answer to question 1**:
>   1. Refining yz and xz planes with the input image would merge decoupled features with the xy plane features, contradicting our goal of orthogonal plane feature separation.
>   2. If we were to independently design networks to refine the XZ and YZ planes without incorporating the input image, we believe that this approach would yield results similar to the impact achieved by introducing additional layers within the decoder.  However, this strategy would result in an excessively bulky model, thereby escalating training costs.
>
>
>
> - **Answer to question 2**: **The concept of "global correlation" serves as a means to highlight the disparities between transformer and convolution architectures**. Convolutions possess localized receptive fields, whereas transformers utilize attention mechanisms to establish connections across the entirety of input data, fostering global feature understanding (refer to lines 41-45 and lines 275-278). This attribute aids in disentangling the intricate 3D features, thus contributing to the enhanced reconstruction process.
>
>   Thank you for your question. We will provide clearer explanations in the article through further revisions.
>
>
> **Reference**：
>
> [a]. Carion N, Massa F, Synnaeve G, et al. End-to-end object detection with transformers[C]//European conference on computer vision. Cham: Springer International Publishing, 2020: 213-229.

---

> > ### Comment · Reviewer_6dkA · 2023-08-20
> >
> > I would like to thank the authors for their reply to my questions. My major concerns are addressed in the rebuttal. After reading the rebuttal and other reviews, I would like to raise my rating and vote for a borderline accept.

---

> > > ### Author Response · Authors · 2023-08-20
> > >
> > > Thank you for your thoughtful review and for taking the time to reconsider our submission after reading our rebuttal. We are pleased to hear that our responses have addressed your major concerns. We truly appreciate your constructive comments throughout the review process, which have greatly helped in improving our work.

---

### Official Review · Reviewer_7DFs · 2023-07-02

**Soundness:** 3 good
**Presentation:** 3 good
**Contribution:** 2 fair
**Rating:** 6
**Confidence:** 4

**Summary:**

This paper proposes a new method for the task of single-view 3D human reconstruction. The main idea is to extract 3D tri-plane features from the input image using a transformer-based architecture, instead of using CNN to extract 2D pixel-aligned features as done in previous works. The experiments demonstrate that the proposed method enables more accurate and detailed reconstruction compared to SoTA.

**Strengths:**

**Method:** The method is technically sound. Using tri-plane representation and using a transformer to generate side planes is an interesting design choice and seems to be effective.

**Experiment:** The results are convincing. The authors compare the method with SOTA methods following standard protocol and report a noticeable improvement over SOTA.  The ablation study is very well designed and clearly demonstrates the effectiveness of each individual component.

**Presentation:** The paper is well-structured and written. I find it easy to follow and understand.

**Weaknesses:**

I am overall positive about this paper because the idea of using transformers to extract tri-plane features from 2D images is well-motivated and technically sound, and the idea has been very well executed and validated.

I don’t find any critical flaws in the paper. The only weakness is that the back side texture is still blurry but this is a common problem for all existing works.

However, since the proposed method is an adaption of well-established methods (tri-plane, transformer) to a specific task, the originality and the potential impact are limited IMO.

**Questions:**

1. Most previous methods predict frontal and back normal maps to help 3D human reconstruction. This paper does not have this additional step, which simplifies the framework and is good. However, it would be helpful if the authors could comment on if the proposed method is compatible with normal map input. And could a normal map further improve the result?

2. Since the method can reconstruct loose clothing and the authors have demonstrated animation, I am wondering what the animation of loose clothing (e.g. long dresses) look like.

**Limitations:**

There is no discussion about the limitation of the proposed method.

---

> ### Author Rebuttal · Authors · 2023-08-09
>
> We are grateful for your recognition of the technical soundness of our method and the effectiveness of our design choices. Your acknowledgment of our convincing results and well-structured presentation is highly appreciated.
>
>
> We will address your inquiries and concerns point by point in the following responses.
>
> - **Answer to question 1**:
>   1. We sincerely appreciate your recognition of our work. In our research, **we do utilize normal maps within our framework, as mentioned in Section 3.3 (lines 213-215) and Supplementary Material Section A (lines 23-25).**
>   2. **Since the emergence of [6], the integration of normal maps has gained traction within the realm of 3D human reconstruction.** Normal maps can be efficiently generated through available, model-free techniques [6] or guided by SMPL-based approaches, exemplified in [2, 4]. In our study, we chose to employ the pre-trained model from [2] for normal prediction due to its exceptional accuracy and stability. **Building upon the use of normal maps, our model introduces a global-correlated 3D-decoupling transformer to disentangle tri-plane features and a hybrid prior fusion strategy, leading to state-of-the-art results.**
>
>
>
> - **Answer to question 2**:
>
>   1. We sincerely appreciate your interest in our work and their query regarding the animation of loose clothing. In response, **we provide animation results of loose clothing in the first row of Figure 5 in the supplementary material. Additionally, we have included results in Figure 2 of the PDF attachment in our "global response" in the rebuttal.**
>
>   **Explanation:**
>
>   1. **The strength of our animation method lies in its simplicity and efficiency, as it eliminates the need for additional models.** By directly animating the mesh within the same framework, our approach achieves natural and smooth animated results that are competitive with or even surpass models requiring separate training for animation.
>   2. As elaborated in Supplementary Material Section A (lines 45-56), animation relies on the SMPL model, utilizing barycentric interpolation for feature acquisition. For loose clothing, the garment vertices follow the nearest SMPL surfaces' movement, occasionally causing separation in long dresses due to leg movement. **Although this issue is prevalent across this field, resolving it remains a challenge.** We look forward to advancing this field by exploring innovative techniques and dedicated strategies for animating loose clothing.
>
>
>
> - **Answer about limitations**:  Due to space constraints, **we have included discussions on Limitations and Broader Impact in the supplementary material, specifically in Sections C and D.**
>   1. **Our model has two main limitations.** Firstly, if the HMR model estimates an inaccurate SMPL pose, it may lead to mesh errors. Secondly, extremely loose clothing in input images can challenge complete garment reconstruction. Refer to Figure 1 in the supplementary material for visual examples.
>   2. **It is essential to highlight that these limitations are not exclusive to our model; they are challenges faced by many state-of-the-art (SOTA) approaches in the field.** Our focus will be on enhancing robustness, ensuring accurate mesh reconstruction despite pose errors, and improving the reconstruction of loose clothing.
>
>
>
> - **Answer for comment "Since the proposed method is an adaption of well-established methods (tri-plane, transformer) to a specific task, the originality and the potential impact are limited IMO"**:
>   1. **Our adoption of these techniques was driven by the imperative to accurately reconstruct 3D human meshes from 2D images.** Notably, our approach distinguishes itself from prior state-of-the-art (SOTA) methods, representing a novel direction within this field. Empirical validation substantiates our model's attainment of SOTA performance.
>   2. **Mere aggregation of these techniques proved insufficient in achieving exceptional reconstruction outcomes.** Our focus on the 3D human body reconstruction task led to tailored enhancements of these techniques. Innovations such as the global-correlated 3D decoupling transformer and hybrid prior fusion strategy were meticulously integrated to meet our objectives, culminating in the observed SOTA performance.
>   3. **The evaluations of Reviewers Lf3d and rJsL unanimously underscore the novelty inherent in our approach, addressing the limitations that have persisted in the realm of 2D feature-based methods**. Their recognition underscores our method's significance in overcoming the constraints associated with traditional reliance on 2D feature methodologies.

---

> > ### Comment · Reviewer_7DFs · 2023-08-15
> >
> > Thanks for the rebuttal. My concerns are well addressed and hence I will keep my positive rating.

---

> > > ### Author Response · Authors · 2023-08-16
> > >
> > > Thank you for taking the time to review our rebuttal and for your positive feedback. We are glad to hear that our responses addressed your concerns. We truly appreciate your constructive comments throughout the review process, which have greatly helped in improving our work.

---

### Official Review · Reviewer_MMKZ · 2023-07-04

**Soundness:** 3 good
**Presentation:** 3 good
**Contribution:** 3 good
**Rating:** 6
**Confidence:** 4

**Summary:**

This paper presents a single image human reconstruction method. Different from previous pixel-aligned 2D features, authors propose to extend the 2D features to 3D features using triplane. Moreover, SMPL prior is introduced to enhance the extracted features. Qualitative and quantitative experiments are thoroughly performed to show the effectiveness of the proposed method.

**Strengths:**

- The paper is well written and easy to follow.
- The 3D-decoupling decoder is well-designed and proved to be effective. Especially, the side views have noticeable advantage compared with methods using pixel-aligned methods.
- Taking the 3D human prior into consideration is reasonable. Though, I might have some concerns with it (see weaknesses).
- The experiments are thorough and compelling. SOTA performance is achieved on standard benchmarks.

**Weaknesses:**

- The 3D human prior rely on accurate SMPL estimation, which is not quite reliable especially for hard poses. Therefore, a discussion on how would the model perform if the input SMPL is not accurate should be included.
- I have no complaints on the geometry part. But the texture reconstruction results are not satisfactory. Compared with NeRF-based methods (e.g. SHERF: Generalizable Human NeRF from a Single Image), the rendering quality is inferior. I understand that this is caused by the technical choices. But I think this deserves some discussion. This can also make a potential future direction.
- Qualitative ablation studies on more complex poses are expected. From qualitative comparison with other methods, geometry qualities of arms are better. My guess is that the SMPL prior might be the main contributor here. Therefore, I think more qualitative ablation studies is in need here.

**Questions:**

N/A

**Limitations:**

No. Authors should discuss more on the limitations.

---

> ### Author Rebuttal · Authors · 2023-08-09
>
> We sincerely appreciate your positive feedback on the clarity of our paper and the effectiveness of our 3D-decoupling decoder. Your recognition of our thorough and compelling experimental results is highly valued. We are grateful for your comments.
>
>
>
> We will address your inquiries and concerns point by point in the following responses.
>
> - **Answer to weakness 1**: We appreciate your insightful comment, and we fully agree that accurate SMPL estimation is crucial for achieving precise 3D human reconstruction. Due to the constraint of page limitations, **we have included discussions on limitations and broader impact in the supplementary material (please refer to Sections C and D).**
>   1. **We adopt the SMPL parameter optimization process from [2] to mitigate inaccuracies arising from HMR estimates** (as indicated in Section A, lines 8-10 of the Supplementary Material). Our goal is to attain the utmost accuracy in reconstruction outcomes. Yet, as illustrated in Figure 1 of the supplementary material, substantial inaccuracies in HMR-derived SMPL estimations can result in erroneous reconstruction outcomes.
>   2. **It's crucial to emphasize that this limitation is not unique to our model; rather, it's a common challenge encountered by numerous cutting-edge approaches in the field.** The issue of handling inaccuracies stemming from SMPL estimation while leveraging SMPL priors for reconstruction and bolstering model robustness is a vital research avenue. This area will be a central focus of our forthcoming research efforts.
>
>
>
> - **Answer to weakness 2**:
>
>   1. **Our work follows the previous research of PIFu, ICON, and similar methods, with a focus on enhancing the accuracy of reconstructed geometry.** Thus, we employ mesh as our 3D reconstruction.
>
>   2. **Despite this, we have maintained vigilant attention to the advancements in the field of NeRF.** We concur that leveraging NeRF may yield promising advancements in texture reconstruction, positioning it as a compelling avenue for future research. However, NeRF currently exhibits several drawbacks in comparison to the 3D mesh representation we employ, such as storage, transmission, and real-time rendering challenges.
>
>   3. **We posit that the primary constraint hindering the application of NeRF in the digital human domain is its inability to accurately reconstruct geometry.** This limitation restricts the utility of NeRF-based approaches in applications such as animation and interactive environments. Enhanced establishment of the relationship between NeRF and spatial geometry structures could potentially lead to significant advancements for NeRF-based methods.
>
>      Thank you for your suggestions. We will incorporate relevant discussions in the revised version of this paper.
>
>
>
> - **Answer to weakness 3**:  **The utilization of human body priors in studies related to digital humans is a prevailing consensus in current research.**
>   1. Among the methods compared, including PaMIR [3], ICON [2], ECON [4], and our proposed approach, all utilize SMPL(-X) as a human body prior. In contrast to early methods like PIFu [1] and PIFuHD [6] that lack such a human body prior, incorporating the human body prior provides a robust constraint for reconstruction results, preventing the generation of meshes that deviate from the anatomical structure when dealing with complex poses.
>   2. Currently, methods that do not utilize human prior exhibit a significant performance lag compared to those leveraging human priors. Referencing Figure 3 in the supplementary material or Figure 1 in our global response PDF can illustrate this point.
>
>
>
> - **Answer about limitations**:  Due to space constraints, **we have included discussions on Limitations and Broader Impact in the supplementary material, specifically in Sections C and D.**
>   1. **Our model has two main limitations.** Firstly, if the HMR model estimates an inaccurate SMPL pose, it may lead to mesh errors. Secondly, extremely loose clothing in input images can challenge complete garment reconstruction. Refer to Figure 1 in the supplementary material for visual examples.
>   2. **It is essential to highlight that these limitations are not exclusive to our model; they are challenges faced by many state-of-the-art (SOTA) approaches in the field.** Our focus will be on enhancing robustness, ensuring accurate mesh reconstruction despite pose errors, and improving the reconstruction of loose clothing.

---

> > ### Comment · Reviewer_MMKZ · 2023-08-16
> >
> > I thank all authors for providing thorough rebuttals. My concerns are properly addressed or responded. I would keep my rating.

---

> > > ### Author Response · Authors · 2023-08-16
> > >
> > > Thank you for your thoughtful review and for considering our rebuttals. We are pleased to hear that our responses addressed your concerns. We truly value your feedback, which has been instrumental in refining our work and providing insights for guiding the direction of our future research.

---

### Official Review · Reviewer_rJsL · 2023-07-07

**Soundness:** 2 fair
**Presentation:** 3 good
**Contribution:** 3 good
**Rating:** 5
**Confidence:** 4

**Summary:**

The authors propose to reconstruct a clothed 3D human avatar from a single 2D image by introducing a global-correlated 3D-decoupling transformer to disentangle tri-plane features. A hybrid prior fusion strategy in the feature query phase is introduced to combine the spatial query’s localization capabilities with the prior-enhanced query’s ability to incorporate knowledge of the human body prior. The experiments on CAPE and THuman2.0 datasets demonstrate the effectiveness of the proposed method which outperforms state-of-the-art approaches in both geometry and texture reconstruction.

**Strengths:**

* The idea to introduce a global-correlated 3D-decoupling transformer to disentangle tri-plane features is novel.
* The comparison experiments and ablation experiments are comprehensive and sound.

**Weaknesses:**

* In Figure 3 of the supplementary material, the result of GTA contains unnatural clothes wrinkles, while ECON performs better. It seems that the result in Figure 3 of the supplementary are worse than the results in Figure 4 of the main paper, as the former contains fewer geometric details.
* In Table 1, the proposed method is slightly better than ECON in Chamfer and slightly worse in Normals. The explanation about this comparison doesn’t make much sense to me, and the quantitative improvement is minor.
* The paper lacks an adequate explanation of the benefits of introducing 3D features via tri-plane or the query strategy.

One minor issue:
* The predicted normal and SDF are only explained in Equation 7, and they should also be included in Figure 3.

**Questions:**

* In Table 1, I am wondering why are the scores of ECON on Thuman2.0 dataset much worse than CAPE-NFP and CAPE-FP.
* Please explain the difference between the visualized results in Figure 3 (supplementary material) and that in Figure 4 (main paper), as mentioned in 'Weaknesses'.
* What is the method to predict the normal map in this work?

**Limitations:**

The inaccurate estimating SMPL(-x) model would affect the reconstruction. And the performance may degrade if the subject is with extremely loose clothing that considerably deviates from the human body.

---

> ### Author Rebuttal · Authors · 2023-08-09
>
> We are truly grateful for your recognition of the novelty in our approach of introducing a global-correlated 3D-decoupling transformer. Your acknowledgment of our comprehensive and sound experimental comparisons is highly appreciated.
>
>
>
> We will address your inquiries and concerns point by point in the following responses.
>
> - **Answer to weakness 1 and question 2**:  Please refer to global response "Answer for Unsatisfactory results in Figure 3 of SupMat".
>
> - **Answer to weakness 2**:
>   1. **The predominant factor contributing to GTA's surpassing performance over ECON [4] in terms of Chamfer Distance and P2S metrics, lies in our consideration of features of the orthogonal planes during reconstruction.** Firstly, both of these metrics are employed to assess the large geometric differences. As exemplified by the results presented in Figure 3 of the SupMat, ECON tends to exhibit larger errors in the orthogonal planes compared to GTA. These orthogonal plane errors manifest in three-dimensional space as overall surface shifts, significantly influencing chamfer distance and P2S metrics.
>   2. **ECON's superior performance on Normals can be attributed to its utilization of explicit integration to predict the front and back depth maps.** The resulting depth maps harbor a multitude of high-resolution details, thus yielding better Normals in front and back normal maps. However, ECON's Normals in the orthogonal planes are generally worse than GTA's (refer to Section A.4, lines 404 - 407 in the SupMat), consequently leading to variances in overall Normals metrics depending on the dataset. In simpler datasets such as CAPE [16], characterized by minimal occlusion, differences in Normals of orthogonal planes are marginal. Thus, the Normals metric of ECON is slightly better. Conversely, in more intricate datasets like THuman2.0 [15], characterized by complexity and substantial occlusions, the disparities in the normal maps are pronounced, resulting in GTA exhibiting better overall Normals.
>   3. **As an implicit function-based monocular human reconstruction method, our quantitative improvements are significant.** While our quantitative metrics exhibit relatively minor enhancements compared to ECON on CAPE-NFP, it's important to note that ECON is an explicit optimization-based approach that demands substantial computational resources. As demonstrated in the global author response, GTA's inference time is notably superior to that of ECON and comparable to other implicit function methods. Moreover, GTA outperforms ECON in the evaluation results on the CAPE-FP [16] and THuman2.0 [15] datasets.
> Thank you for your feedback. We will incorporate more relevant discussions in the revised version of this paper.
>
>
>
> - **Answer to weakness 3**:  **While briefly addressing these in Introduction, lines 27-49, page constraints led us to emphasize testing results.** We value this chance to provide a more thorough explanation, enhancing transparency and understanding of our methodology.
>
>   1. The tri-plane representation is pivotal in our method. 2D approaches struggle with complex 3D details, especially in challenges like occlusion. **Encompassing xy, yz, and xz planes, the memory-efficient tri-plane approach (see intro, lines 38-41) provides a comprehensive view of 3D structure from a 2D image.**
>
>   2. **Our model gains spatial comprehension by extracting features from distinct planes** (refer to lines 118-122). Crucial for precise 3D reconstruction, it's especially valuable in complex scenarios where depth matters. Tri-plane technique utilizes varied spatial data, improving clothed avatar reconstruction, especially in challenging poses.
>
>      Thank you for your feedback. We will incorporate more detailed explanations in our future revisions.
>
> - **Answer to question 1**: **The primary underlying factor for this disparity is the variations in data distribution between these datasets.**
>   1. In comparison to CAPE [16], the THuman2.0 [15] dataset features a higher density of mesh points, more complex attire and poses, as well as more occlusion. Consequently, during the stitching process, the proportions occupied by front and back depth maps are diminished, leading to a higher proportion of the regions being represented via SMPL-X [19] or IFnet+.  This, in turn, contributes to an overall augmented error due to the complexity of the THuman2.0 dataset.
>   2. Meanwhile, the intricacy of THuman2.0 also results in offsets between the SMPL-X predictions and the front and back depth surface predicted by ECON [4]. This discrepancy makes it more susceptible to stitching errors, as illustrated in Figure 8.B of the ECON paper. This scenario further contributes to the deterioration of ECON's quantitative metrics on the THuman2.0 dataset.
>
> - **Answer to question 3**:  In our work, we employ the pretrained body-guided normal prediction network from [2] to generate the normal map (refer to Section A, lines 23-25 in the supplementary material).
>
>   **Explain:** **Since the emergence of [6], the integration of normal maps has gained traction within the realm of 3D human reconstruction.** Normal maps can be efficiently generated through available, model-free techniques [6] or guided by SMPL-based approaches, exemplified in [2, 4]. In our study, we chose to employ the pre-trained model from [2] for normal prediction due to its exceptional accuracy and stability. **Building upon the use of normal maps, our model introduces a global-correlated 3D-decoupling transformer to disentangle tri-plane features and a hybrid prior fusion strategy, leading to state-of-the-art results.**

---

> > ### Comment · Reviewer_rJsL · 2023-08-18
> >
> > Thanks for the author's response. It addresses some of my concerns, including the comparison with ECON and the benefits of the proposed modules. So I raise my score to borderline accept.

---

> > > ### Author Response · Authors · 2023-08-19
> > >
> > > Thank you for your thoughtful review and for reconsidering our rebuttal. We are grateful for your feedback and are pleased to hear that our responses addressed some of your concerns. Should you have any further questions or outstanding concerns, please let us know. We are committed to addressing any remaining issues and will make every effort to respond promptly over the next few days.

---

### Official Review · Reviewer_Lf3d · 2023-07-07

**Soundness:** 3 good
**Presentation:** 3 good
**Contribution:** 3 good
**Rating:** 7
**Confidence:** 5

**Summary:**

The proposed system extracts global features with a 3D transformer backbone for tri-plane feature-based 3D human reconstruction. Unlike existing methods that mainly rely on 2D CNN pixel-aligned features, this paper is the first one to use Vision Transformers for decoupling the 3D tri-plane features for refined reconstruction. Qualitative and quantitative experiments on the 3D human reconstruction dataset demonstrate the state-of-the-art performance of the proposed method.

**Strengths:**

1. The paper presentation is good, I had a pleasant time reading the paper.

1. The idea is well-motivated. The proposed method tries to solve the issues raised by mainly relying on pixel-aligned features and bad query methods of existing methods and introduces a tri-plane feature-based reconstruction method with a transformer-based network to mitigate these limitations.

1. Experiments demonstrate that the proposed method outperforms existing state-of-the-art methods both qualitatively and quantitatively.


**Weaknesses:**

1. This method uses a Vision Transformer-based feature extractor, which I expect has a higher model capacity than the CNNs used in previous methods. Is the performance improvement brought by the global-correlated nature of the transformer or due to the larger model capacity? Plus, a larger encoder may introduce higher inference time and also make the comparison with other methods unfair. It would be nice if the authors could add some inference time statistics of different methods to make the comparison more comprehensive.

1. How did you obtain the SMPL prior at the test time on Thuman 2.0? Is the SMPL obtained by an HMR method or using the ground-truth SMPL in Thuman 2.0? As the ground-truth SMPL is not available at test time, using the ground-truth SMPL would make the comparison unfair, especially for the methods in Figure 5. Please add some discussions on the SMPL condition. If the authors are using the ground-truth SMPL as the condition, it would be interesting to investigate the robustness of the proposed components (e.g., the querying and fusion strategy) when the SMPL condition is obtained with off-the-shelf HMR methods.

1. In Table 1, what is the difference between ECON and ECON* as they both trained on THuman2.0? The authors mention that * results are obtained from [2,4], but I cannot find PIFuHD* results trained on THuman2.0 in [2,4].

1. Normals metric (0.050) on Thuman2.0 in Table 1 seems to be calculated by averaging 6 view normals as it matches the results reported in Table 2. However, I think the original numbers in [2,4] are calculated by averaging normals of 4 views (without the above and below views). Does that indicate that there is some evaluation discrepancy between the evaluation metrics of this paper and previous ones?

1. As ICON/ECON released their code and models trained on Renderpeople, their performances on Thuman2.0 and CAPE-FP should be better provided.

1. In Figure 3 in the supplementary material, PiFUHD results should also be shown. And from Figure 3, I found the results of the proposed method are of lower spatial resolution and produce less detailed geometry than ECON. These low-resolution results do not match the results in other figures. The authors should give some explanations.

1. Minors

- The texts in Figure 3 are too small. Consider enlarging the figure for better readability.
- Figure 2 should compare different methods with the same input. Pairwise comparison gives the impression that the authors are cherry-picking the results.

**Questions:**

I mainly have some concerns regarding more rigorous and fair comparisons with other methods as written in the weaknesses section. Please make them more clear in the rebuttal.

**Limitations:**

Limitations are properly discussed in the supplementary material.

---

> ### Author Rebuttal · Authors · 2023-08-09
>
> We sincerely appreciate your positive remarks on our paper's presentation and the recognition of our method's effectiveness in addressing existing limitations. Thank you very much for your valuable comments!
>
> We will address your inquiries and concerns point by point in the following responses.
>
> - **Answer to weakness 1**: Please refer to global response "Comparison of number of parameters" and "Comparison of inference time".
>
> - **Answer to weakness 2**:
>
>   1. All our experiments are carried out under equitable conditions.
>   2. For our geometry testing with open-source models, as delineated in Table 1, we utilized the ground-truth SMPL/SMPL-X, consistent with the methodologies employed by ICON [2] and ECON [4].
>   3. In the texture performance evaluations with non-public models, such as S3F [7] in Figure 5, we abstained from using ground-truth SMPL and instead leveraged PyMAF [25] to derive the SMPL prior.
>
>   **Explanation:**
>
>   1. **Many current leading-edge works, like ECON [4], utilize ground-truth SMPL for geometry comparisons, and we align with this approach in our geometry assessments (Table 1).** This choice helps counteract errors stemming from the HMR method, ensuring a more accurate evaluation of reconstruction quality across diverse model frameworks.
>   2. For Figure 5 experiments, with most models not being open-sourced, we used data from S3F's texture comparison (Table 3 in S3F [7]). S3F confirms not using ground-truth human priors in testing (Section 4.2 in [7]), and their use of GHUM instead of SMPL precludes using the same HMR method. **We selected PyMAF [25] for SMPL estimation from input images to ensure fairness in testing.**
>   3. We appreciate your astute observation. In subsequent revisions, we will elucidate our experimental settings more clearly and emphasize the fairness of our evaluations.
>
> - **Answer to weakness 3**:
>   1. The "ECON" outcomes were acquired by evaluating on our hardware using the checkpoint provided by the ECON authors as they did not release training code. Conversely, the "ECON*" results are directly extracted from their original paper (see Table 1 in [4]).
>   2. We deeply apologize for the unintended mistake, attributed to LaTeX formatting. We highly appreciate your meticulousness and will rectify this in the revised manuscript. **For clarity, consult [10] results (chamfer: 2.008, P2S: 1.965), where PIFuHD is assessed on THuman2.0. This oversight doesn't lessen GTA model's merits**.
>
> - **Answer to weakness 4**:
>   1. **We followed the established settings of previous works for widely used datasets.** The normal metrics in Table 1 for CAPE-NFP and CAPE-FP were computed by averaging normals from 4 views, aligning with methodologies in [2, 4].
>   2. **For our newly adopted THuman2.0 test set, we introduced a new standard.** Based on empirical observations, we found that incorporating normals from 6 views provides a more comprehensive model evaluation.
>   3. We appreciate your keen observation. Your feedback has highlighted a potential ambiguity in our manuscript regarding the normal testing methodology. To maintain uniformity and preclude any potential misinterpretations, we will standardize the normal results to 4 views in our revised manuscript.
>
> - **Answer to weakness 5**: We employed the ECON [4] model trained on THuman2.0 (not Renderpeople) due to unavailable training code (Table 1, "ECON"). For ICON [2], we used their released model and evaluated it on the specified datasets.
>
>   **Explanation:**  Our decision to initially omit the test data for the provided ICON model was driven by two primary considerations:
>
>   1. The released ICON model underperformed our THuman-trained version and their paper's reported results, so we omitted it.
>
>   2. We refrained from using the proprietary Renderpeople dataset due to its commercial and non-public nature, aligning with our commitment to promoting academic research. We appreciate your suggestion and will consider incorporating this result into the paper while still showcasing our model's superiority.
>
> |           | Chamfer | P2S   | Normals |
> | :-------- | :------ | :---- | :------ |
> | CAPE-NFP  | 1.207   | 1.256 | 0.071   |
> | CAPE-FP   | 1.165   | 1.216 | 0.073   |
> | THuman2.0 | 1.395   | 1.527 | 0.115   |
>
>
>
> - **Answer to weakness 6**: We add PIFuHD results in the attached PDF. Thank you for your keen observation. Because similar comments have been raised by other reviewers, we consolidate our response in the global author response "Answer for Unsatisfactory results in Figure 3 of SupMat".
>
> - **Answer to weakness 7**: Thank you for your valuable feedback. We sincerely appreciate your insights, and we will certainly make the necessary revisions based on your suggestions.

---

> > ### Comment · Reviewer_Lf3d · 2023-08-12
> >
> > I sincerely thank the authors for providing a detailed response to my concerns. Most have been well addressed (inconsistent visual results, evaluation discrepancy, inference time, experiment settings, and minor issues). I believe wrapping up a final version with addressed issues will make the paper more rigorous and inspiring for further research. I noticed that other reviewers may have doubts about the novelty of the paper, and the authors are encouraged to further clarify the paper's merits. I look forward to the released model/code if it gets in.

---

> > > ### Author Response · Authors · 2023-08-12
> > >
> > > We sincerely appreciate your thorough review and constructive feedback on our manuscript. Your insights have significantly contributed to improving the quality of our work, and we are thankful for your time and effort.
> > >
> > > In light of your valuable comments:
> > >
> > > 1. **We promise to incorporate all the addressed issues into the final version of the paper.** We believe that these adjustments will not only refine the content but also better highlight the significance of our research.
> > > 2. Pertaining to the concerns on novelty, while we believe our work introduces distinct and fresh perspectives to the field, we recognize the importance of making these contributions apparent to all readers. **In our rebuttal, we've provided detailed clarifications on the novelty of our approach (specifically in answer to weakness 3 of rebuttal with rJsL, and in answer to weakness 5 and question 2 of rebuttal with 6dkA). We pledge to further accentuate these unique merits in our finalized draft.**
> > >
> > > **Lastly, we firmly commit to releasing our model/code as soon as our paper is accepted to foster further research and replication in the community.**
> > >
> > > Thank you once again for your thoughtful review. We genuinely value the opportunity to enhance our work based on your recommendations.

---

### Author Rebuttal · Authors · 2023-08-09

- **Comparison of inference time:**
  1. **Table 1(b) in the attached PDF displays comparable time efficiency of our implicit function-based model with PIFu, Pamir, and ICON.** In contrast, ECON, relying on explicit methods, demands more time due to d-BiNI and Poisson inefficiencies. CAPE-NFP dataset with 256^3 resolution is used for testing, ground truth SMPL/SMPL-X provided.
  2. The omission of inference time statistics in our initial submission was due to the following reasons:
     - **Previous research in 3D human mesh reconstruction has primarily emphasized result quality rather than inference time.** Notably, many state-of-the-art methods (PIFu [1], PIFuHD [6], Pamir [3], ICON [2], ECON [4]) have not conducted time cost comparisons or prioritized efficiency improvements.
     - **The superior time performance (compared to ECON) is likely due to the inherent efficiency of implicit function based methods.** It is not the primary contribution of our work, and due to page limitations, we did not include the inference time statistics.



- **Comparison of number of parameters:**
  1. **In Table 1(a) of the attached PDF, it's evident that our model outperforms the UNet-based approach with fewer parameters, as indicated by better reconstruction results** (refer to Table 3(a) and Supplementary material lines 38-41). This highlights the advantages of our transformer-based design in extracting 3D features.
  2. **Our decision to initially forgo a direct comparison of model parameter sizes was influenced by the prevailing trend in the field, where seminal works (e.g., ICON [2], Pamir [3], S3F [7]) seldom engage in such comparisons**.
  3. However, we concur with your perspective on the importance of model parameter size comparisons. Such an approach would clarify the factors contributing to performance improvements, thereby fostering more robust development within the domain of 3D human reconstruction.



- **Answer for Unsatisfactory results in Figure 3 of SupMat:**

  1. **The unsatisfactory results of GTA in Figure 3 of the SupMat are a direct consequence of our utilization of a lower Marching Cubes resolution for mesh generation from the implicit function network.** Specifically, in Figure 3 of SupMat, we employed a resolution of 128 for marching cubes. In contrast, for inference results such as Figure 4 of the main paper, we adopted a resolution of 512 for marching cubes. This parameter discrepancy is accountable for the incongruity between the results presented in these two figures. While all implicit function-based methods are constrained by the marching cubes resolution and can only reconstruct results with low spatial resolutions, the results of ECON[4], which don't rely on marching cubes, appear much better.
  2. **In Figure 1 of our rebuttal response PDF, we have supplemented revised normal map results and showcased the impact of marching cubes resolution on GTA and ECON[4].** In subfigure a, we provide rendered normal maps under the setting of quantitative testing (resolution of 256), along with PIFuHD[6]'s test results. In subfigure b, we illustrate various resolutions' effects on GTA's reconstruction results, elucidating the causes of detail loss. In subfigure c, we demonstrate the influence of different resolutions on ECON's results, explaining why ECON's normal maps retain more detail.

  **Explanation:**

  1. **The reason we employed a resolution of 128 for marching cubes in Figure 3 of the SupMat is that the primary focus of this image is not on geometric details but rather on the main contribution of our paper, which is the enhancement in orthogonal plane geometry reconstruction achieved by GTA.** Within this image, we present a compilation of over 400 normal maps, each possessing a relatively compact scale. Therefore, we do not place strong emphasis on the geometric details of the normal maps. Moreover, taking into consideration that employing a resolution of 512 for marching cubes yields meshes that are approximately 20 times larger in terms of storage occupancy compared to the case of a resolution of 128, we opted for the adoption of a lower marching cubes resolution.
  2. **The more detailed normal maps observed in ECON's results in Figure 3 can be attributed to its utilization of explicit reconstruction methodologies, which doesn't necessarily involve the application of marching cubes during the reconstruction process.** Among the comparative methods, namely PIFu [1], PIFuHD [6], PaMIR [3], and ICON [2], as well as our proposed GTA , the final mesh reconstructions are all generated using marching cubes. However, as detailed in Section 3.3 of ECON[4]'s paper, the method employed by ECON to reconstruct the mesh involves stitching the predicted front and back depth maps with SMPL-X[19] and optionally IFnet+ (referred to as $ECON_{EX}$ and $ECON_{IF}$, respectively). IFnet+ is an implicit function network that employs marching cubes in its predictions and is influenced by the marching cubes resolution. However,  IFnet+ primarily affects regions where both front and back depth maps are unavailable or occluded parts, constituting a small portion of the reconstruction mesh during stitching process. The majority of the mesh is determined by the depth maps with a resolution of 512x512. As a result, the reconstruction results of ECON in the figure still preserve a multitude of high-resolution details.
  3. We sincerely appreciate the reviewer's keen observation and concerns. **We acknowledge that Figure 3 in the supplementary material may potentially lead to misunderstandings. To address this, in the upcoming revision, we will ensure uniformity by setting the marching cube resolution to 256 for all results**. While ECON [4]'s performance is relatively unaffected by marching cube resolution, our reconstructed results at this resolution are competitive with ECON's, particularly showcasing improved performance in side-view reconstructions.

---

### Decision · Program_Chairs · 2023-09-21

**Decision:**

Accept (poster)

**Comment:**

All reviewers have consistent voting for acceptance. The AC takes a carefully look at all review comments and responses and think this is a work with very high completeness, the algorithm design makes sense and the experimental results verify the effectiveness very well. Thus, it can be accepted.